# Anthropogenic forcings reverse a simulated multi-century naturally-forced Northern Hemisphere Hadley cell intensification

Or Hess [1] ✉ & Rei Chemke [1]

The Hadley circulation plays a central role in determining the location and intensity of the hydrological cycle in tropical and subtropical latitudes. Thus, the human-induced historical and projected weakening of the Northern Hemisphere Hadley circulation has considerable societal impacts. Yet, it is currently unknown how unparalleled this weakening is relative to the response of the circulation to natural forcings in past centuries. Here, using state-of-the-art climate models, we show that in contrast to the recent and future human-induced Hadley circulation weakening, natural forcings acted to intensify the circulation by cooling the climate over the last millennium. The reversal of a naturally-forced multi-centennial trend by human emissions highlights their unprecedented impacts on the atmospheric circulation. Given the amplifying effect of natural forcings on the Hadley circulation, our analysis stresses the importance of adequately incorporating natural forcings in climate model projections to better constrain future tropical climate changes.

The Hadley Circulation (HC) is largely responsible for transferring heat and moisture between tropical and subtropical latitudes in both hemispheres[1]. In addition, the vertical motions within the HC set the distribution of precipitation over those regions. Specifically, the HC is composed of an ascending branch of moist and warm air near the equator, which produces high precipitation rates in the deep tropics. In the upper troposphere, the air propagates poleward to the subtropics, where a descending branch of dry air sets the position of the arid zones on Earth (e.g., the subtropical deserts) before returning equatorward near the surface. Thus, even modest changes in the HC could considerably impact the regional climate in tropical and subtropical latitudes[2-5].

By the end of this century, in response to anthropogenic emissions, climate models project with high confidence a weakening of the HC only in the Northern Hemisphere (NH)[6,7], which was argued to be mainly driven by changes in the atmospheric temperature structure[6-9]. In contrast to climate models, observation-based atmospheric reanalyses generally show an intensification of the circulation in recent decades, which has been suggested to be an artifact[10-12] that could stem from biases in latent heating[10]. Although the above conclusions were based on precipitation observations (rather than surface

easterlies wind observations[13]), which are uncertain over the ocean[14,15], recently, by employing sea-level pressure measurements as a proxy for the NH HC strength, it was found that the NH HC has been weakening in recent decades, consistent with climate models, and such weakening has been attributed to anthropogenic emissions[16].

While recent and future changes in the strength of the NH HC have been extensively studied, it has yet to be quantified how unparalleled these changes are relative to the NH HC changes over past centuries. Knowledge of the climate in past centuries would not only allow one to assess how unprecedented is human-induced climate change but also to improve our understanding of climate change induced by natural forcings (relative to natural forcings, anthropogenic forcings had a weaker influence on the pre-industrial climate[17]). A well-known such analysis is the "hockey stick" figure[18], which emphasizes the degree of influence of anthropogenic emissions on NH surface temperature. Unlike surface temperature, the lack of observed wind records limits the ability to examine the HC changes in past centuries[19]. Nevertheless, the reconstruction of past external forcings allows one to use climate model simulations to assess the response of the climate to external forcings over the last millennium (defined here as the preindustrial 850-1849 period). For example, relative to the last millennium, climate

[1]Department of Earth and Planetary Sciences, Weizmann Institute of Science, Rehovot, Israel. ✉e-mail: or.hess@weizmann.ac.il

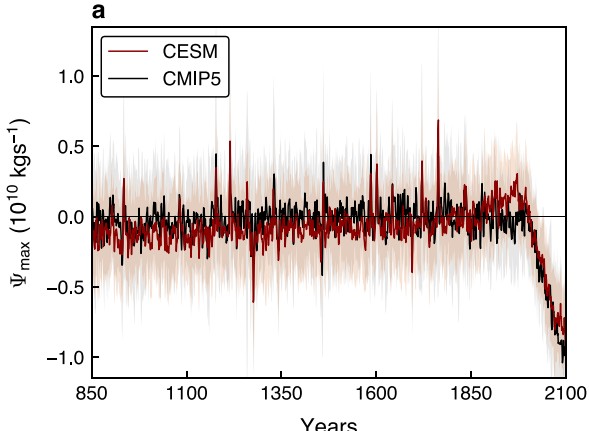
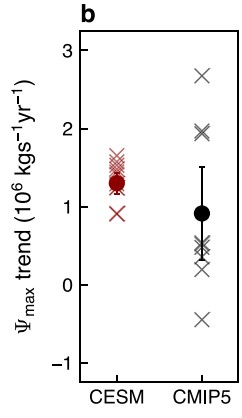

**Fig. 1 | The evolution of the Northern Hemisphere Hadley cell strength.**
**a** Evolution of the Northern Hemisphere Hadley cell strength ($\Psi_{max}$), relative to the 1810-1850 period, in the Community Earth System Model (CESM) mean (red line) and Coupled Model Intercomparison Project Phase 5 (CMIP5) mean (black line). Shading shows standard deviation across members/models. The evolution has been smoothed with a 3-year running mean for plotting purposes. The evolution of individual members/models is shown in Supplementary Fig. 12. **b** The 850-1849 $\Psi_{max}$ trends in CESM (red) and CMIP5 (black). The red and black dots show the CESM and CMIP5 mean trends, respectively, and cross the individual members/ models. Error bars show the 95% confidence interval of the mean trend based on a Student's $t$ distribution.

models simulate an unprecedented poleward displacement of zonal surface winds over the 20th century[20].

The aim of this work is thus to quantitatively contextualize the current and future weakening of the NH HC in response to anthropogenic emissions with respect to the NH HC's changes over the last millennium (we here focus on the NH HC since anthropogenic emissions have a minor impact on the circulation in the Southern Hemisphere[7]). In addition, since climate model projections do not fully account for future natural forcings[21] (e.g., changes in volcanic forcings are not taken into account in climate model projections[22–24]), investigating the impacts of natural forcings on the NH HC over the last millennium would allow one to better constrain climate change projections in the tropics to subtropics.

## Results
### Unprecedented human-induced weakening of the Hadley cell
To assess how unprecedented is the weakening of the NH HC in response to anthropogenic emissions, we start by examining the evolution of the NH HC strength ($\Psi_{max}$, Methods) over the 850-2100 period. Specifically, over the last millennium, we analyze nine models that participate in Phase 5 of the Coupled Model Intercomparison Project (CMIP5, Methods) as well as the 12-member Community Earth System Model Last Millennium Ensemble (CESM-LME, Methods), under the last millennium forcings (Methods). To complement the time series through 2100, we analyze the same CMIP5 models and the Community Earth System Model Large Ensemble (CESM-LE, Methods) under the historical (through 2005) and Representative Concentration Pathway 8.5 (RCP8.5; through 2100) experiments (Methods). Here, as noted in previous studies[25,26], we focus on the NH HC changes in the mean of the ensembles (CMIP5 or CESM), which allows us to isolate the forced response of the NH HC to external forcings (anthropogenic and/or natural) from the internal climate variability (Methods).

Examining the NH HC strength evolution over the 850-2100 period (Fig. 1a) reveals that both in CESM mean (red line) and CMIP5 mean (black line), the human-induced weakening of the circulation in recent and coming decades is unprecedented compared to the NH HC forced changes over the last millennium (in accordance with recent work on the observed NH HC weakening[16], the forced weakening of the NH HC in recent decades is attributed to anthropogenic emissions, Methods, Supplementary Fig. 1). Quantitatively, using a signal-to-noise ratio

approach[26,27], we estimate the year when the anthropogenically forced weakening of the NH HC emerges out of the forced NH HC changes over the last millennium. The signal is defined as the ensembles mean $\Psi_{max}$ trends of different lengths, all starting at 1970 (the year when the ensembles mean $\Psi_{max}$ starts to weaken), and the noise as the mean plus one standard deviation (s.d.) of all trends with the same length as the signal that occur during the 850-1849 period in the ensembles mean. The time of emergence is defined as the year when the signal exceeds 2 s.d. (relative to the mean) of forced NH HC changes in response to last millennium forcings, i.e., signal-to-noise ratio smaller than -2. We find that in both CESM and CMIP5 mean, by the early 00s (2010-2020), the human-induced $\Psi_{max}$ weakening trend emerged out of the last millennium forced NH HC changes (Fig. 2a; similar results are also evident under a milder warming scenario over the 21st century, Methods, Supplementary Fig. 2).

Interestingly, not only that the human-induced weakening of the NH HC is unprecedented since 850, but it also reverses a multicentennial strengthening of the flow (Fig. 1; similar results are also evident in CMIP6 models, Methods, Supplementary Fig. 3). Using linear regression analysis, over the last millennium (the 850-1849 period), the NH HC exhibited a statistically significant intensification at a rate of $1.3 \times 10^6$ kg$^{-1}$ yr$^{-1}$ in CESM mean and at a rate of $0.9 \times 10^6$ kg$^{-1}$ yr$^{-1}$ in CMIP5 mean (Fig. 1b; we note that such intensification is insensitive to the chosen starting and end years for the regression, Methods). Note that in comparison to the spread in $\Psi_{max}$ trends across CESM members (which represents the internal variability of NH HC trends[25], Methods), the spread across CMIP5 models is larger, suggesting that the different models' configurations might also affect the magnitude of the NH HC strengthening over the last millennium. Nevertheless, all CESM members and almost all CMIP5 models (except for one) simulate a strengthening of the circulation over the 850-1849 period.

By employing an attribution analysis, the forced intensification of the flow is found to be mostly attributed to natural forcings—the major contributors to climate changes in the last millennium (Methods, Supplementary Fig. 4). This is noteworthy considering that prior to 1850, human activities (e.g., deforestation, agriculture, and population growth) impacted global climate by influencing surface albedo and the biogeochemical cycle[28–31]. However, anthropogenic forcing alone could not adequately explain this rate of circulation intensification, and its contribution is considerably smaller than that of natural forcings.

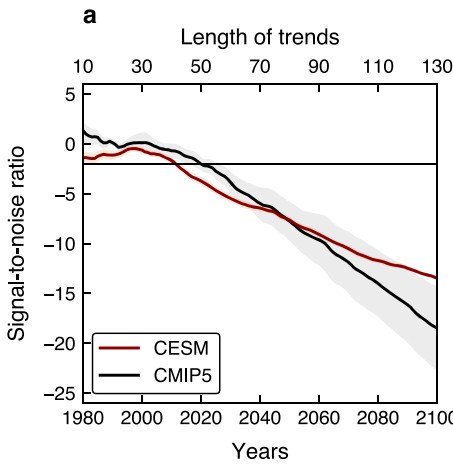
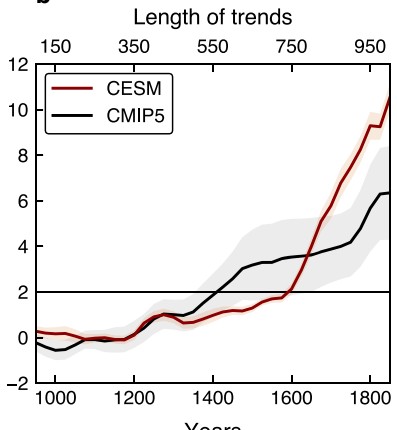

**Fig. 2 | Time of emergence of the forced response of the Hadley cell strength.** Signal-to-noise ratio analysis to the Northern Hemisphere Hadley cell strength trend from (**a**), 1970 and (**b**), 850 to each year plotted against the last year of trend in the Community Earth System Model (CESM) mean (red line) and Coupled Model Intercomparison Project Phase 5 (CMIP5) mean (black line). Shading shows the standard deviation of signal-to-noise ratio values (Methods). The horizontal black line represents a signal-to-noise ratio value of -2 in (**a**) and value of 2 in (**b**). The evolution has been smoothed with a 3-year running mean for plotting purposes.

Lastly, to estimate the degree of influence of natural forcings on the NH HC over the last millennium, we next ask: does the naturally forced NH HC intensification emerged out of the internal climate variability? To answer this question, we again use a signal-to-noise ratio approach, where the signal is defined as trends of different lengths since 850 (the first year of the last millennium simulations) to each year (until 1849) and the noise as s.d. of all trends, with the same length as the signal, that arise only due to internal variability (estimated from preindustrial control runs[26], Methods).

In CESM mean (red line in Fig. 2b), the intensification of the NH HC emerged from the internal climate variability around 1600, and in CMIP5 mean around 1400 (black line). Moreover, the intensification emerged between 1500–1700 in all CESM members and between 1350-1750 in most CMIP5 models (except for two). The larger spread in time of emergence across CMIP5 models, relative to the spread across CESM members, again highlights that the different models' configurations might impact the magnitude of the NH HC strengthening trend. Nonetheless, the emergence of the intensification in both CESM and CMIP5 mean, all CESM members and almost all CMIP5 models not only stress how significant was the naturally forced strengthening of the NH HC over the last millennium, but also suggests that the strengthening of the circulation in individual runs can be mostly attributed to natural forcings; internal variability alone could not adequately explain the NH HC intensification.

## The source of the naturally induced Hadley cell strengthening

Opposite trends over the last millennium and over the 20th and 21st centuries are not only evident in the NH HC strength but were also found in the NH surface temperature[32,33]. In contrast to the ongoing and future surface warming in the NH, over the last millennium, between the Medieval Climate Anomaly (MCA, 950-1250) and Little Ice Age (LIA, 1450-1850), cooling of the surface in the NH was observed, which was argued to stem from changes in natural forcings[34–37]. It is thus conceivable that while under human-induced global warming the NH HC weakens[38,39], under naturally induced global cooling it strengthens.

To examine the above hypothesis, we focus on $\Delta\Psi_{max}$ (where $\Delta$ is the difference between the LIA and MCA) for consistency with the extensive work on last millennium climate changes; we note that $\Delta\Psi_{max}$ yields similar results as we find in the linear trend over the entire last millennium (Supplementary Fig. 5). Specifically, we use the Kuo-Eliassen (KE) equation (Methods) to isolate and quantify the mechanisms that contributed to the naturally forced strengthening of the NH

HC between the MCA and LIA. The KE equation, which was shown to adequately capture the recent and future changes in the NH HC strength[7,10], is an elliptic linear diagnostic equation that takes the following simple form: $L\Psi = D_Q + D_{v'T'} + D_{u'v'} + D_X$, where $L$ is a second-order elliptic operator, which is a function of the Coriolis parameter and static stability ($S^2$). In addition to the operator, on the right-hand side (RHS) there are four processes that also affect the HC strength including: diabatic heating ($D_Q$), which can be further decomposed into contributions from latent ($Q_{lat}$) and radiative ($Q_{rad}$) heating, meridional eddy heat ($D_{v'T'}$) and momentum ($D_{u'v'}$) fluxes and zonal friction ($D_X$) (Methods). We here analyze the KE equation using CESM (and not CMIP5) since, unlike CMIP5, CESM has available output data of eddy fluxes and diabatic heating components in the last millennium runs; yet, we verify the CESM results in CMIP5 models.

Before exploiting the KE equation to understand via which processes natural forcings acted to intensify the circulation over the last millennium, we first ensure that the changes in the NH HC strength between the MCA and LIA, as inferred from solving the KE equation ($\Delta\Psi_{max}^{KE}$), well captures the changes in $\Psi_{max}$ ($\Delta\Psi_{max}$). Indeed, $\Delta\Psi_{max}^{KE}$ is highly correlated with $\Delta\Psi_{max}$ ($r = 0.97$, Fig. 3a) with a slight overestimation by the KE equation, which likely stems from the quasi-geostrophic assumptions used in deriving the equation[40]. This, together with the linearity of the KE equation, provide us the confidence to examine the relative roles of each term in the KE equation in strengthening the NH HC (Methods).

We find that the terms that contribute the most to the intensification of the NH HC over the last millennium are the meridional gradient of latent heating ($Q_{lat}$) and static stability ($S^2$) (Fig. 3b). All other terms have minor weakening contributions to the NH HC strength. Interestingly, the human-induced NH HC weakening in recent and coming decades was also found to be dominated mostly by changes in static stability and latent heating[7,10].

## Linking the Hadley cell strengthening to the cooling over the last millennium

How do the changes in static stability and latent heating result in the strengthening of the NH HC between the MCA and the LIA? To answer this question, we examine the zonal mean atmospheric temperature changes over the last millennium (Fig. 4a). Similar to the overall cooling of the surface[32,33], the troposphere cooled between the MCA and LIA, but not uniformly across latitudes and levels. Specifically, at low latitudes, the upper troposphere cooled more than the lower troposphere, as one would expect following changes in the moist

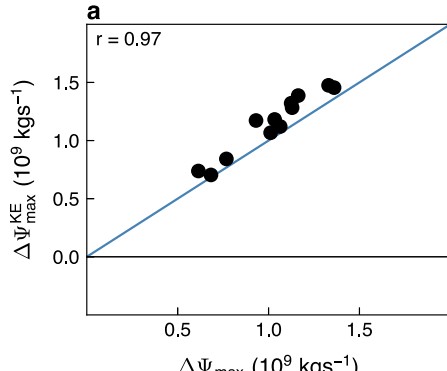
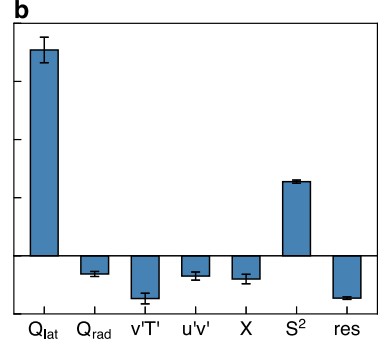

**Fig. 3 | The source of the Hadley cell strengthening over the last millennium.** **a** The difference in the Northern Hemisphere Hadley cell strength between the Little Ice Age (LIA) and the Medieval Climate Anomaly (MCA) in the Community Earth System Model (CESM) as inferred from the Kuo–Eliassen (KE) equation ($\Delta\Psi_{max}^{KE}$) as a function of $\Delta\Psi_{max}$; their correlation appears in the upper left corner.

The blue line represents the $y = x$ line. **b** The relative contribution to $\Delta\Psi_{max}^{KE}$ from latent heating ($Q_{lat}$), radiative heating ($Q_{rad}$), eddy heat flux ($v'T'$), eddy momentum flux ($u'v'$), zonal friction ($X$), static stability ($S^2$) and the residual. Error bars show the 95% confidence interval based on a Student's t-distribution.

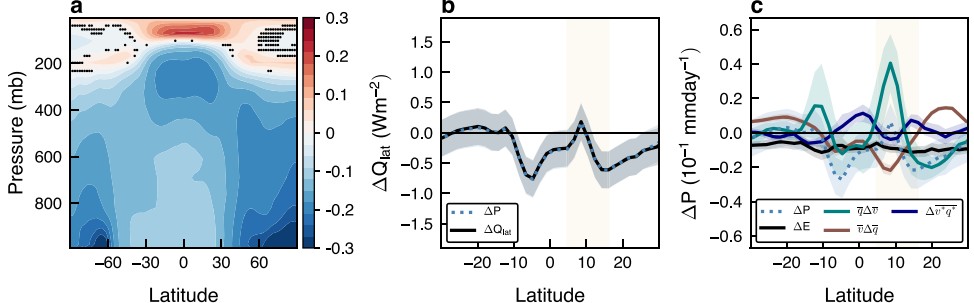

**Fig. 4 | The role of static stability and latent heating in the Hadley cell changes.** The difference between the Little Ice Age (LIA) and Medieval Climate Anomaly (MCA) of the zonal mean (**a**), temperature (K) and (**b**), precipitation ($\Delta P$, blue) and latent heating ($\Delta Q_{lat}$, black) in the Community Earth System Model (CESM) mean. The black dots in panel a show where less than two-thirds of the members agree on the sign of change. **c**, The contribution of the different terms in the moisture

budget equation to changes in precipitation: mean meridional circulation ($\overline{q}\Delta\overline{v}$, green line), mean moisture ($\overline{v}\Delta\overline{q}$, brown line), eddy moisture flux ($\Delta\overline{v^*q^*}$, purple line) and evaporation ($\Delta E$, black line). Shading shows standard deviation across members. The orange shading shows the Northern Hemisphere Hadley cell ascending branch region.

adiabatic lapse rate under surface cooling (a similar cooling pattern appears in CMIP5 models, Supplementary Fig. 6a). This cooling pattern acts to destabilize the troposphere (reduce static stability) and, thus, to intensify the NH HC over the last millennium. We note that static stability plays an opposite role under surface warming over the 20th and 21st centuries when it is found to result in the weakening of the NH HC[7,8]. The similar tropospheric temperature patterns only of opposite signs over the last millennium and over the 21st century (Supplementary Fig. 7) result in the opposite impacts of static stability on the NH HC strength.

Next, to link latent heating changes to the intensification of the NH HC, we examine the changes in the vertically integrated latent heating between the MCA and LIA (black line in Fig. 4b). Over the climatological ascending branch of the NH HC (average over the MCA period), which is defined between the latitude of the Inter Tropical Convergence Zone (ITCZ, where $\Psi$ at 500 mb changes sign between the SH and NH HCs) and the latitude of $\Psi_{max}$[7] (orange shading in Fig. 4b), the meridional gradient of latent heating increased over the last millennium. According to the KE equation, this increase resulted in the strengthening of the NH HC (similar results appear in CMIP5 models, Supplementary Fig. 6b). To further understand these changes in latent heating, we next analyze the changes in surface precipitation between the MCA and LIA since the net latent heating in an atmospheric column equals surface precipitation (compare blue and black lines in Fig. 4b). Specifically, we follow previous studies[3,41–43] and

examine the changes in the zonal mean vertically integrated moisture budget equation (Fig. 4c, Methods), which takes the following simple form: $\Delta\overline{P} = \left[\overline{q}\Delta\overline{v} + \overline{v}\Delta\overline{q} + \Delta\overline{v^*q^*}\right]_y + \Delta\overline{E}$, where square brackets represent vertical integral and meridional convergence, overbar is zonal and annual mean, asterisk is deviation from zonal and annual mean, $P$ is precipitation, $q$ is specific humidity, $v$ is meridional velocity and $E$ is evaporation. The first two terms on the RHS ($\overline{q}\Delta\overline{v}$, $\overline{v}\Delta\overline{q}$) respectively represent the contributions from changes in mean meridional circulation and mean specific humidity, the third term ($\Delta\overline{v^*q^*}$) represents the change in eddy moisture flux, and the fourth term ($\Delta E$) is the change in evaporation.

We find that, over the NH HC ascending branch, changes in the mean circulation ($\overline{q}\Delta\overline{v}$) contributed the most to a positive peak and the associated increase in the meridional gradient in precipitation (and in latent heating) (green line in Fig. 4c). All other components have minor or opposite effects on these precipitation changes. Thus, in addition to our comprehension that a decrease in static stability intensified the NH HC, the above analysis indicates a profound interaction between the naturally forced HC intensification and the meridional gradient of latent heating over the HC ascending branch. One possible explanation for this interaction could be that the NH HC intensification over the last millennium might have acted as positive feedback, i.e., the increase in $\Psi_{max}$ increased latent heat release over the ascending branch of the NH HC, resulting in a larger meridional gradient of latent heating, which

further strengthened $\Psi_{max}$. However, given that casual relationships cannot be drawn from the above equations, further investigation is imperative to furnish sufficient evidence to categorize the interaction between the NH HC and latent heating over the last millennium.

Interestingly, despite the opposite temperature responses to anthropogenic emissions in coming decades (i.e., warming) and to natural forcings over the last millennium (i.e., cooling) (Supplementary Fig. 7), an increase in the meridional gradient of latent heating also acts to intensify the NH HC by the end of this century[7]. In future warming scenarios, thermodynamic arguments (i.e., 'the wet gets wetter, dry gets drier' argument[42]) explain the increase in precipitation and in its meridional gradient in the deep tropics. Over the last millennium, on the other hand, the increase in precipitation and in its meridional gradient stems from dynamical changes (Fig. 4c), while thermodynamic arguments cannot explain an increase in precipitation; according to the 'wet get wetter, dry get drier' mechanism, under a reduction in temperature (as occurs between the MCA and LIA), one would expect to obtain a reduction in precipitation in regions of moisture flux convergence (e.g., over the ascending branch of the NH HC). Thus, the similar tendency of latent heating to intensify the NH HC both over the last millennium and over the 21$^{st}$ century, in spite of the opposite temperature changes, stems from the different effects of thermodynamic and dynamic processes in driving these latent heating changes.

Lastly, it is conceivable that changes in the NH HC strength might also be linked to cross-equatorial energy transport due to uneven hemispherical heating induced by natural forcings and accompanied changes in ITCZ location via modulation of the meridional gradient of latent heating[44]. However, similar to the climate response over the 21$^{st}$ century[45], not only does the ITCZ position and width exhibit minor changes over the last millennium (Supplementary Figs. 8, 9) but also the Northern and Southern Hemispheres low latitudes exhibit similar changes in temperatures over the last millennium (Fig. 4a).

## Discussion

To stress the human influence on the climate system, the latest IPCC report shows that human-induced thermodynamic changes, such as the warming of the surface and the increase in sea-level rise, are unprecedented with respect to past centuries[46]. However, there is large uncertainty in how unprecedented are human-induced large-scale dynamical changes. Here, using state-of-the-art climate models, we examine the evolution of the NH HC strength over the 850-2100 period and find that the anthropogenically forced NH HC weakening in recent and coming decades is unprecedented relative to the naturally forced NH HC changes during the preindustrial last millennium. Moreover, we find that in contrast to the human-induced warming over the 20$^{th}$ and 21$^{st}$ centuries, the cooling induced by natural forcings over the last millennium (between the MCA and LIA) acted to intensify the NH HC in past centuries. Thus, not only that the human-induced weakening of the NH HC is unprecedented, but it also reverses a multi-century naturally forced strengthening trend of the NH HC. Our results not only highlight the unequivocal impact of human emissions on the atmospheric circulation, but given the large impacts of the HC on the hydrological cycle[1], the unprecedented changes in the NH HC could also cascade to regional climate impacts at low to subtropical latitudes. We note that similar to the minor impact of anthropogenic emissions on the Southern Hemisphere HC strength over the 20$^{th}$ and 21$^{st}$ centuries, natural forcings also have minor effects on the strength of the flow in the Southern Hemisphere over the last millennium (Supplementary Fig. 10).

Lastly, the significant NH HC strengthening along the preindustrial last millennium and our additional analysis on the physical mechanism underlying it emphasize that natural forcings affected the circulation over the last millennium. Hence, not adequately incorporating natural forcings in climate model projections reduces the

accuracy of climate-change adaption and mitigation strategies. This study underscores the importance of further investigating how natural forcings have been and will influence atmospheric dynamics alongside the existing research on the impacts of anthropogenic forcings.

## Methods
### The Hadley cell strength
The strength of the NH HC ($\Psi_{max}$) is defined, following previous studies[5,16,47], as the maximum value, at 500 mb, of the meridional mass stream function ($\Psi$),

$$\Psi(\phi,p) = \frac{2\pi a \cos\phi}{g} \int_0^p \bar{v}(\phi,p')\mathrm{d}p', \qquad (1)$$

where $a$ is Earth's radius, $g$ is gravity, $\phi$ is latitude, $p$ is pressure, $v$ is the meridional velocity, the overbar represents a zonal and annual mean, and $p'$ is the pressure variable of integration. We choose to analyze $\Psi_{max}$ as a metric for the HC strength since the NH HC intensification over the last millennium is evident throughout most of the NH HC (Supplementary Fig. 11). Consequently, analyzing a spatially averaged metric for the HC strength[48,49] ($\Psi_{avg}$, the averaged streamfunction between 10° and 25° and between 1000 mb and 100 mb) yields similar results; the unprecedented human-induced weakening of the HC reverses a multi-centennial forced strengthening of the circulation (Supplementary Figs. 12–14). In addition, we choose to analyze the annual mean NH HC strength for consistency with the discussion in the latest IPCC report[46], and since the NH HC is projected to weaken in all seasons[6,10,50,51].

### CMIP5 models
We use monthly output from nine CMIP5 models[52] (CCSM4, FGOALS-S2, GISS-E2-R, HADCM3, IPSL-CM5A-LR, MIROC-ESM, MPI-ESM-P, MRI-CGCM3, and BCC-CSM1-1) forced between 850-1849 under the last millennium forcings (part of Phase 3 of the Palaeoclimate Modelling Intercomparison Project, PMIP3[53]), between 1850-2005 under the historical forcings and between 2006-2100 under the RCP8.5 scenario (to weigh all models equally we use all models with available output of meridional winds under the 'r1i1p1' member). We also make use of five models with available outputs under milder warming scenarios, i.e., the RCP4.5 scenario (GISS-E2-R, IPSL-CM5A-LR, MIROC-ESM, MPI-ESM-P, and MRI-CGCM3). We here choose to analyze in the main text CMIP5 data rather than CMIP6 data, as only three CMIP6 models have available output for the analysis over the last millennium (EC-EARTH3-VEG-LR, MIROC-ES2L, and MRI-ESM2-0), and thus cannot be used to estimate the forced response in the system. Nevertheless, using CMIP6 models yields the same main conclusion revealed by CMIP5 models, which is that the human-induced weakening of the NH HC reverses a multi-centennial strengthening of the circulation (Supplementary Fig. 3). In addition, we focus here on the last millennium since not only it is adjacent to the industrial era, but the vast array of proxy records (e.g., tree rings, ice cores, boreholes, etc.) over this period provides a good reconstruction of external forcing agents, and thus places the last millennium as one of the ideal periods for studying the response of the climate system to natural forcings. To estimate the internal variability in the system (the noise) in the detection of the naturally forced NH HC strengthening over the 850-1849 period, we use long preindustrial control runs forced with a constant 1850 forcings[26] (total of 4900 years of unforced data).

### CESM-LME simulations
The CESM-LME is the largest single-model ensemble of simulations under the last millennium forcings (12 members)[54]. Each member in the CESM-LME is initialized with slightly different conditions, which yields distinct climate evolution in each member due to the chaotic nature of the system. Thus, while the CESM-LME mean allows assessing the forced response in the system to external forcings (anthropogenic

and/or natural), the spread across the CESM members accounts for the internal climate variability[25]. We analyze the CESM-LME between 850 and 1919 under the same forcings as in CMIP5. We note that the size of the CESM-LME ensemble is sufficiently large to capture the variability and forced response of the NH HC intensity; an ensemble size of eight members already captures 95% of the NH HC strength variability (Supplementary Fig. 15). Lastly, similar to the analysis in CMIP5 models, we make use of an 850-control run (~1350 years forced with a constant 850 forcings which branched from the 1850 control simulation) for estimating the internal variability in the detection analysis for the naturally forced NH HC strengthening.

## CESM-LE simulations

To extend the NH HC evolution in CESM-LME to 2100 and examine the anthropogenically forced circulation changes, we use 12 members (randomly selected out of 40) of the CESM-LE[55] integrated between 1920-2100, under the CMIP5 historical forcings and the RCP8.5 future scenario. Similar to CESM-LME, the use of a single-model large ensemble allows us to disentangle the internal climate variability from the climate's forced response to external forcings. We here select 12 CESM members (the same number of members as in CESM-LME) to equally assess the forced response of the NH HC to external forcings over the last millennium and over the 20th and 21st centuries. We further ensure that our results are independent on which 12 members are randomly selected; repeating the time of emergence analysis for the human-induced NH HC weakening using 100 different sets of 12 randomly selected members yields similar results of emergence in the early 00s of the forced weakening of the NH HC out of the naturally forced NH HC changes in the last millennium.

## Evaluation of the long-term forced response

To evaluate the forced response signal over the last millennium, we employ a linear regression analysis. To further validate the robustness of our results, we also employ the Mann–Kendall test[56] (Supplementary Fig. 16), which yields a consistent significant strengthening of the NH HC in the ensemble means, as was found using the linear regression analysis. Additionally, we find that the forced response HC intensification is insensitive to the chosen time interval[57] (Supplementary Fig. 17). Robust forced strengthening trends are found across various time intervals, spanning from 600 to 1000 years, with widespread agreement among individual members/models.

## Attribution analysis

To assess whether the forced simulated NH HC weakening in recent decades can be attributed to anthropogenic emissions, we make use of natural-only forcing runs in CMIP5 and CMIP6 (hist-nat, same as the historical runs but including only natural forcings). Here we use all models with available data under the hist-nat experiment in CMIP6 (ACCESS-ESM1-5, BCC-CSM2-MR, CANESM5, CESM2, FGOALS-G3, GFDL-ESM4, IPSL-CM6A-LR, MIROC6, MRI-ESM2-0, NORESM2-LM) and CMIP5 (CCSM4, CESM1-CAM5, CNRM-CM5, CSIRO-Mk3-6-0, CANESM2, FGOALS-G2, GFDL-CM3, GFDL-ESM2M, GISS-E2-H, GISS-E2-R, IPSL-CM5A-MR, MIROC-ESM, MIROC-ESM-CHEM, MRI-CGCM3, BCC-CSM1-1). Specifically, the forced NH HC trend in recent decades under the historical runs (where both anthropogenic and natural forcings are present) is compared to the trend under the hist-nat runs (where only natural forcings are present) (Supplementary Fig. 1). In recent decades, the relatively small trend under the hist-nat runs, in comparison to the historical runs, suggests that the forced NH HC weakening can be attributed to anthropogenic emissions.

Similarly, to attribute the forced NH HC strengthening over the last millennium to natural forcings, we make use of 5 CESM-LME single-forcing ensembles[54]. In each ensemble (of 3-5 members), a different natural (solar flux, volcanic eruptions, greenhouse gases, and orbital changes) or anthropogenic (land-use land-cover forcings) forcing

agent freely evolves throughout the 850-1849 period, while all other forcing agents are held fixed. The contribution of each forcing agent to the NH HC changes is estimated as the NH HC 850-1849 trend in the mean of each single forcing ensemble. Then, the overall impact of natural forcings is estimated by combining the contribution from each natural forcing agent (Supplementary Fig. 4). All natural forcings agents combined capture most of the NH HC strengthening over the last millennium (as simulated by CESM-LME), suggesting that this strengthening can be mostly attributed to natural forcings. Since the single forcing ensembles comprise different numbers of members, which could affect the variability in each ensemble, we estimate the forced response (the mean across the members) after applying a bootstrapping sampling procedure (of 10 iterations) across the ensemble members, thus creating the same size of single forcing ensembles. We here choose to analyze CESM data rather than CMIP5 data since, unlike CMIP5 models, CESM has available output for single-forcing ensembles over the last millennium.

## Detection analysis

We use a signal-to-noise ratio analysis for the detection of the anthropogenically or naturally forced responses of the NH HC strength. The definitions of the signal and noise are given in the main text for each detection analysis. We note that detrending the pre-industrial control runs prior to estimating the noise for the detection of the naturally forced NH HC intensification over the last millennium yields the same results. Lastly, to estimate the uncertainty in the detection analyses we apply a bootstrap sampling procedure (of 100 iterations) to the CESM members/CMIP5 models used to calculate the signal. The uncertainty in the signal-to-noise ratio is then defined as s.d. of signal-to-noise ratio values across the bootstrapping iterations.

## The Kuo-Eliassen equation

Following previous work[7,10,51], we use the Kuo-Eliassen (KE) equation to elucidate the physical processes that contributed to the intensification of the NH HC between the MCA and LIA. The KE equation is an elliptic linear diagnostic equation derived using quasigeostrophic approximations and thermal wind balance[58,59], which in spherical coordinates takes the following form

$$
\begin{aligned}
& f^2 \frac{g}{2\pi a \cos\phi} \frac{\partial^2 \Psi}{\partial p^2} + S^2 \frac{g}{2\pi a} \frac{\partial}{a\partial\phi} \frac{1}{a\cos\phi} \frac{\partial \Psi}{\partial\phi} \\
& = \frac{R}{p}\left( \frac{1}{a}\frac{\partial \overline{Q}}{\partial\phi} - \frac{\partial}{a\partial\phi}\frac{1}{a\cos\phi}\frac{\partial \overline{v'T'}\cos\phi}{\partial\phi} \right) \\
& + f\left( \frac{1}{a\cos^2\phi}\frac{\partial^2 \overline{u'v'}\cos^2\phi}{\partial p\partial\phi} - \frac{\partial \overline{X}}{\partial p} \right),
\end{aligned} \tag{2}
$$

where $f$ is the Coriolis parameter, $S^2 = -\frac{1}{\overline{\rho}\overline{\theta}}\frac{\partial\overline{\theta}}{\partial p}$ the static stability, $\overline{\rho}$ the density, $\overline{\theta}$ the potential temperature, $R = 287 \mathrm{Jkg}^{-1}\mathrm{K}^{-1}$ (the gas constant of dry air), $\overline{Q}$ the diabatic heating, $\overline{v'T'}$ and $\overline{u'v'}$ the eddy heat and momentum fluxes, respectively, where prime represents a deviation from zonal and monthly mean and $\overline{X}$ the zonal friction, estimated using the annual and zonal mean zonal momentum quasi-geostrophic equation. The eddy fluxes and diabatic heating components are available outputs of the CESM model.

The linearity of the KE equation allows us to estimate the relative contribution of each term in Eq. (2) to changes in $\Psi_{max}^{KE}$. Specifically, by rewriting Eq. (2) as

$$
L\Psi = D, \tag{3}
$$

where $L$ is the linear operator on the LHS, and $D$ is the sum of the terms on the RHS, and decomposing $L$, $\Psi$, and $D$ into their MCA values and

deviation from that period[10,60], $L = L_{MCA} + \Delta L$, $\Psi = \Psi_{MCA} + \Delta\Psi$ and $D = D_{MCA} + \Delta D$, Eq. (3) yields an equation for the change in the HC strength relative to the MCA period ($\Delta\Psi$),

$$L_{MCA}\Delta\Psi = \Delta D - \Delta L\Psi_{MCA} - \Delta L\Delta\Psi, \qquad (4)$$

where $\Delta D$ represents changes in each of the RHS terms in Eq. (2), $\Delta L\Psi_{MCA}$ represents the changes in static stability (the operator), and $\Delta L\Delta\Psi$ represents the multiplicative changes in static stability and the streamfunction (calculated as a residual). Note that since we analyzed the annual mean streamfunction, the terms in Eq. (4) are nonindependent.

## Moisture budget

We investigate the vertically integrated annual mean moisture budget equation[41] to isolate and quantify the physical mechanisms underlying the surface precipitation changes[3,42,43]:

$$\Delta\overline{P} = -\frac{1}{g}\nabla_y \int_0^{p_s} (\Delta\overline{vq})dp + \Delta\overline{E}, \qquad (5)$$

where $P$ is precipitation, $p_s$ is surface pressure, $q$ is specific humidity and $E$ is surface evaporation. By decomposing the moisture flux ($\overline{vq}$) to its mean and eddy components ($\overline{vq} = \overline{v}\,\overline{q} + \overline{v^*q^*}$, where asterisk represents a deviation from zonal and annual mean), Eq. (5) can be rewritten as follows,

$$\Delta\overline{P} = -\frac{1}{g}\nabla_y \int_0^{p_s} \left([\overline{q}]\Delta\overline{v} + [\overline{v}]\Delta\overline{q} + \Delta\overline{v^*q^*}\right)dp + \Delta\overline{E}, \qquad (6)$$

where the square brackets represent the mean value of the MCA and LIA. The RHS terms represent the contributions to precipitation changes from changes in moisture ($\Delta\overline{q}$), mean circulation ($\Delta\overline{v}$), eddy flux ($\Delta\overline{v^*q^*}$) and evaporation ($\Delta\overline{E}$). Here we define the eddies ($\overline{v^*q^*}$) as deviation from zonal and annual mean to account for long-term disturbances in the tropics (e.g., the Madden-Julian Oscillation). Using deviations from monthly mean yields similar results.

## Data availability

The data used in the manuscript is publicly available for CMIP5 (https://esgf-node.llnl.gov/projects/cmip5/), CMIP6 (https://esgf-node.llnl.gov/projects/cmip6/) and CESM (https://www.earthsystemgrid.org/).

## Code availability

The code for calculating the KE equation is available at https://doi.org/10.5281/zenodo.7954604.

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

## Acknowledgements

O.H. is supported by the Israeli Science Foundation Grant 906/21. R.C. is grateful to the support by the Willner Family Leadership Institute for the Weizmann Institute of Science.

## Author contributions

O.H. and R.C. downloaded the data, O.H. analyzed the data, and together, they discussed and wrote the paper.

## Competing interests

The authors declare no competing interests.
