## [Peer Review File · Nature Communications]

Anthropogenic forcings reverse a simulated multi-century naturally-forced Northern Hemisphere Hadley cell intensificationReviewers' comments:

Reviewer #1 (Remarks to the Author):

General comments:

My main concern is the novelty of this work. First, Chemke and Polvani (2021), their Fig. 1a, and Chemke and Polvani (2019), their Figs. 1cd, have employed the same methodology and already shown that the magnitude of current NH HC weakening is unprecedented, though only by showing the time series from 1900 to 2100. The net radiative forcing has not changed much from year 850 to 1900 in comparison to the magnitude of the changes in 1900-2100 period, so the Hadley circulation is also not expected to change much too. The results of this study are therefore expected. Second, the same methodology of employing KE equation has been applied to multiple studies before and can be repeated on a bunch of other datasets too.

I have also several methodological concerns. First, the authors have not included CMIP6 simulations, which use a more advanced PMIP4 forcings instead of PMIP3. They argue in lines 234-236: "We here choose to analyse CMIP5 data rather than CMIP6 data, as only two CMIP6 models have available output for the analysis over the last millennium." Second, they constrain their CMIP5 input data even more by using only those, which output meridional winds (lines 233-234). While I understand such choice as the authors employed stream-function to estimate the Hadley cell intensity, note that there exists a plethora of other metrics of Hadley cell intensity and these can be used instead of stream function to provide a more robust analysis of Hadley circulation changes.

Third, the authors opted to use RCP5-8.5 scenario as a proxy for "anthropogenic forcings". I would here suggest using RCP4.5 or SSP2-4.5 if they include the CMIP6. These two scenarios are most aligned with current and projected anthropogenic forcings.

I was surprised by the inaccuracy of the opening statement ("The Hadley Circulation (HC) is largely responsible for transferring energy (in the form of heat and moisture) from low to subtropical latitudes in both hemispheres.") The Hadley circulation barely transports moisture poleward ("from low to subtropical latitudes"), as most of the moisture in the isolated convective towers rapidly condenses and falls out as precipitation in the Tropics. The lack of poleward moisture transport can be observed from time- and zonal-mean meridional transport of specific humidity ($v \partial q / \partial y$). The bulk of moisture transport in the Hadley circulation is equatorward in the lower branches of the Hadley cells. In relative terms, i.e. if the transport is normalised by the amount of humidity in a certain grid box ($v/q \partial q / \partial y$), the upper-tropospheric moisture transport actually becomes significant, but does not meaningfully shape the planetary climate due to low absolute amount of moisture in the upper-troposphere.

I am also not fully convinced by the methodology of attributing NH HC strengthening to natural sources, described in lines 283-291. It would be helpful if the authors would provide some mathematical description of what they did. If I understand correctly, the authors did 4 forcing experiments with an ensemble. Let's denote the natural forcings f_1, f_2, f_3, f_4 . In each of the experiments, one of the

forcing terms, say f_i , was allowed to evolve and the others were kept constant, $f_i=f_i(t)$ and $f_{(j \neq i)}=f_j(0)$, where $t=0$ indicates the starting year 850. The sum of all forcings should be $f=f_1+f_2+f_3+f_4+f_a$, where f_a denotes anthropogenic forcing. My first question is what happened to the anthropogenic forcings present in PMIP3? Was the term f_a allowed to evolve over time or was it kept fixed in all experiments (or was it even evolved in all experiments)? This essential part of methodology is missing making it very unclear. The error bars in Ext.Fig.1c for the “Natural only” experiments are large. The relative contribution from each natural forcing should be shown separately including the anthropogenic forcing. How is the relative contribution defined – describe mathematically!

Throughout the paper, the climate variability before the industrial age, i.e. in the past centuries, is exchanged for natural variability. For example, in lines 42-44, the authors wrote “Nevertheless, climate model simulations of the last millennium allow one to assess the response of the climate to natural forcings in recent centuries.” Similar lines of text can be found in the Methodology, e.g. “To estimate the internal variability in the system (the noise) in the detection of the naturally-forced HC strengthening over the 850-1849 period, ...”

Note that prior to industrial revolution, humans affected the global climate by land-use change (deforestation, agriculture) and the population boom (the population quintupled between year 800 and 1850), affecting the surface albedo and the biogeochemical cycle (Pongratz, 2008; Pongratz, 2009; Pongratz and Caldeira, 2012; Doughty, 2013). This is accounted for in PMIP3 and stated in the literature cited by the authors (Schmidt et al., 2011).

In scope of that, the title “Anthropogenic forcings reverse a multi-century naturally-forced Hadley cell intensification” is inaccurate too: 0) it should be explicitly mentioned that the title is valid only for northern hemisphere Hadley cell; 1) the multi-century changes prior to industrial revolution were not entirely natural as might be implied from the title and 2) the title implies that the Hadley cell intensification in the last millennium is a fact, while no comparison with paleoclimate data is provided or reference to other studies. This is particularly strange as the authors (lines 236-239) justify the use of last millennium forcing by the availability of “vast array of proxy records (e.g., tree rings, ice cores, boreholes, etc.) over this period” which “places it as one of the ideal periods for studying the response of the climate system to natural forcings.”

Some statements in the text should also be modified, for example the one in 31-33: “In spite of the biases found in reanalyses, which resulted in an artificial strengthening of the circulation in recent decades, the weakening of the circulation, as simulated by models, has been observed in recent decades and attributed to anthropogenic emissions.”

While the source of reanalyses biases are both biased observations and biased models, the study of Chemke and Polvani (2019) did not provide a strong evidence of reanalyses biases, despite the claims. Their conclusion was based on a comparison between reanalyses precipitation and GPCP precipitation. Note that the GPCP precipitation data over the oceans are a product of satellite retrieval techniques, which in large part use the same raw data as the reanalyses. As such, the GPCP data over the oceans are prone to large uncertainties, as shown in Prakash et al. (2013), and later by Good et al. (2021, their Fig. 1). Taking GPCP observations as a truth for reanalyses verification, particularly over the oceans, is a very questionable approach.

There are also positives which should be mentioned. The main goal of this paper is interesting in my opinion, but the analysis should be extended, and possibly complemented with the use of paleoclimate data. The detection analysis is performed robustly, and the moisture budget analysis is very informative too.

However, based on the several major points above, I find the paper unsuitable for publication in Nature Communications.

Specific comments:

21: too simplified: the global HC consists of many regional cells with distinct characteristics, which govern the zonal and meridional distribution of precipitation

23: near the equator (ITCZ specifically)

30-33: The biases in the reanalyses that occur in the data assimilation procedure due to biased (satellite) observations are undisputed. These can be only verified by comparing reanalysis products to trustworthy conventional observations, e.g. Simmons (2022).

The biases also affect the intensity of Hadley circulation, among others. However, note that some studies have suggested that these biases are complemented by the natural long-term atmospheric variability, namely Atlantic Multidecadal Oscillation (AMO), which could also temporarily oppose the ongoing weakening of the Hadley circulation (Zaplotnik et al., 2022). The hint to the link between the AMO and HC has been given by Baines et al., 2007, and others. Presenting the recent Hadley circulation strengthening in the reanalyses as a sole consequence of biases is therefore rather inaccurate.

44-45: What does “an unprecedented increase in the magnitude of the global pattern of surface winds over the 20th century” mean? In particular, I am puzzled by the wording of “the magnitude of the global pattern”. How do you define a magnitude of a pattern? This part needs to be rewritten in a less dubious way.

38, 43 and 47: as stated in the Major comments’ section, the forcings in the last millennium before 1850 were not only natural.

221-223: the increase in intensity is very inhomogeneous over the NH HC in Ext. Fig. 4b for CMIP5 mean. Note that precisely for this reason, Nguyen et al. (2013), Pikovnik et al. (2022) and others have derived metrics of average HC intensity by employing either vertical or spatial averaging.

219: gravity acceleration = gravitational acceleration + centrifugal force

248-249: the spread across the members is only a proxy for current-state internal climate variability. The internal climate variability can itself attain a new state after climate undergoes forced response, bringing it into another state.

254: “850-control run”. Is that control run initiated in year 850?

283: NH HC

References:

Baines, P. G., and C. K. Folland, 2007: Evidence for a Rapid Global Climate Shift across the Late 1960s. *J. Climate*, 20, 2721–2744, <https://doi.org/10.1175/JCLI4177.1>.

Braconnot et al, The Paleoclimate Modeling Intercomparison Project contribution to CMIP5, CLIVAR Exchanges No. 56, Vol. 16, No.2, May 2011, pp 15-19

Chemke, R., Polvani, L.M. Opposite tropical circulation trends in climate models and in reanalyses. *Nat. Geosci.* 12, 528–532 (2019). <https://doi.org/10.1038/s41561-019-0383-x>

Chemke, R., & Polvani, L. M. (2021). Elucidating the mechanisms responsible for Hadley cell weakening under $4 \times \text{CO}_2$ forcing. *Geophysical Research Letters*, 48, e2020GL090348. <https://doi.org/10.1029/2020GL090348>

Christopher E. Doughty (2013) Preindustrial Human Impacts on Global and Regional Environment. *Annual Review of Environment and Resources* 2013 38:1, 503-527

Doughty, Christopher E., Preindustrial Human Impacts on Global and Regional Environment (October 2013). *Annual Review of Environment and Resources*, Vol. 38, pp. 503-527, 2013, Available at SSRN: <https://ssrn.com/abstract=2343669> or <http://dx.doi.org/10.1146/annurev-environ-032012-095147>

Good, P., Chadwick, R., Holloway, C.E. et al. High sensitivity of tropical precipitation to local sea surface temperature. *Nature* 589, 408–414 (2021). <https://doi.org/10.1038/s41586-020-2887-3>

Nguyen, H., A. Evans, C. Lucas, I. Smith, and B. Timbal, 2013: The Hadley Circulation in Reanalyses: Climatology, Variability, and Change. *J. Climate*, 26, 3357–3376, <https://doi.org/10.1175/JCLI-D-12-00224.1>.

Pikovnik, M., Zaplotnik, Ž., Boljka, L., and Žagar, N.: Metrics of the Hadley circulation strength and associated circulation trends, *Weather Clim. Dynam.*, 3, 625–644, <https://doi.org/10.5194/wcd-3-625-2022>, 2022.

Pongratz, J. and Caldeira, K. 2012 *Environ. Res. Lett.* 7 034001

Pongratz J, Reick C, Raddatz T and Claussen M 2008 A reconstruction of global agricultural areas and land cover for the last millennium *Glob. Biogeochem. Cycles* 22 GB3018

Pongratz J, Reick C, Raddatz T and Claussen M 2009 Effects of anthropogenic land cover change on the carbon cycle of the last millennium *Glob. Biogeochem. Cycles* 23 GB4001

Satya Prakash, C. Mahesh & R. M. Gairola (2013) Comparison of TRMM Multi-satellite Precipitation Analysis (TMPA)-3B43 version 6 and 7 products with rain gauge data from ocean buoys, *Remote Sensing Letters*, 4:7, 677-685, DOI: 10.1080/2150704X.2013.783248

Schmidt, G. A., Jungclaus, J. H., Ammann, C. M., Bard, E., Braconnot, P., Crowley, T. J., Delaygue, G., Joos, F., Krivova, N. A., Muscheler, R., Otto-Bliesner, B. L., Pongratz, J., Shindell, D. T., Solanki, S. K., Steinhilber, F., and Vieira, L. E. A.: Climate forcing reconstructions for use in PMIP simulations of the last millennium (v1.0), *Geosci. Model Dev.*, 4, 33–45, <https://doi.org/10.5194/gmd-4-33-2011>, 2011.

Simmons, A. J.: Trends in the tropospheric general circulation from 1979 to 2022, *Weather Clim. Dynam.*, 3, 777–809, <https://doi.org/10.5194/wcd-3-777-2022>, 2022.

Zaplotnik, Ž., M. Pikovnik, and L. Boljka, 2022: Recent Hadley Circulation Strengthening: A Trend or Multidecadal Variability?. *J. Climate*, 35, 4157–4176, <https://doi.org/10.1175/JCLI-D-21-0204.1>.

Reviewer #2 (Remarks to the Author):

Review

This study newly presents a Hadley cell strengthening in the last millennium (850 ~ 1850 year), which is driven by natural forcing-induced cooling. The authors convincingly show the increasing trend of the Hadley cell by employing CMIP and large-ensemble experiments. Additionally, their application of theoretical approaches provides valuable understanding of the various contributions to this trend. The scientific findings presented here are of considerable interest not only to the tropical climate community but also to researchers in the historical science field.

However, I think the current manuscript is not substantive enough to warrant publication in Nature communications in the current form. The main concern I have is the need for a more comprehensive explanation of Northern Hemisphere (NH) Hadley cell changes. As such, I have included several comments below, based on my reading of the manuscript. Please review the provided feedback for further consideration.

Major comments

1. As suggested by the authors in L117-118, it may not be surprising that global cooling induces the strengthened Hadley circulation in the linear thinking of substantial research on the weakening of the Hadley cell under global warming (e.g., Lau and Kim 2015; Lu et al. 2008). Also, as suggested in L150-154, the tropical cooling (following the wet adiabat) reduces the static stability, which is exactly the opposite of what happens in global warming scenarios (Figure 3b,d in Chemke and Polvani 2021).

It is clear that the natural forcing of the last millennium regulates the Hadley cell through the Q, which is the biggest difference to global warming. Although the authors show that the Q response is mainly driven by mean circulation changes, the suggested feedback is too vague to understand the Hadley cell response.

For me, it looks like there is a clear southward shift of the ITCZ, especially in the CMIP (Extended Data Figure 4). So it is very plausible that the dynamical feedback is related to the ITCZ. Then, the Hadley cell response over the last millennium could not be directly compared with that of global warming, considering its relatively symmetric changes (e.g., Figure 1 in Lau and Kim 2015). As previous studies already show that the interhemispheric energy difference could induce the ITCZ response (e.g., Kang 2020), I think further analysis to converge the authors' theory and previous findings on the ITCZ response to the thermal forcing is needed and really helpful to understand the Hadley cell dynamics. Or, at least, detail explanation and discussion on the dynamic feedback is necessarily included.

2. In line with comment 1, it looks like the authors are using the NH HC changes and the HC changes without any clarification, which makes me quite confused. I think that because the ITCZ shift can induce asymmetric changes between the two Hadley cells, NH HC might not be enough to represent the overall Hadley cell changes. However, for example, L174-177 discuss the ascending branch of the HC related to the two cells, where they only consider NH

HC ascending branch. Also, the discussion generally talks about the HC changes rather than NH HC changes, although it cannot simply explain the SH HC changes. Please provide solid reasons for considering NH HC as representative of the general tropical circulation. Without this, clarification of the overall statement would be helpful.

Minor comments

1. L79: it looks like a typo 'the early 00s'
2. L113-117: Considering the large intermodel spread in the time of emergence, is it appropriate to compare just two periods rather than employing linear trend over the last millennium?
3. L157: Definition of NH HC ascending branch should be noted in the main manuscript.
4. Figure 3a: including $y=x$ line would be helpful.
5. Figure 4a: I do not see any reason why the Hadley response should not be shown in this figure.

Chemke, R., and L. M. Polvani, 2021: Elucidating the Mechanisms Responsible for Hadley Cell Weakening Under $4 \times \text{CO}_2$ Forcing. *Geophys. Res. Lett.*, **48**, e2020GL090348, <https://doi.org/10.1029/2020GL090348>.

Kang, S. M., 2020: Extratropical Influence on the Tropical Rainfall Distribution. *Curr. Clim. Change Rep.*, **6**, 24–36, <https://doi.org/10.1007/s40641-020-00154-y>.

Lau, W. K. M., and K.-M. Kim, 2015: Robust Hadley Circulation changes and increasing global dryness due to CO_2 warming from CMIP5 model projections. *Proc. Natl. Acad. Sci.*, **112**, 3630–3635, <https://doi.org/10.1073/pnas.1418682112>.

Lu, J., G. Chen, and D. M. W. Frierson, 2008: Response of the Zonal Mean Atmospheric Circulation to El Niño versus Global Warming. *J. Clim.*, **21**, 5835–5851, <https://doi.org/10.1175/2008JCLI2200.1>.

Reviewer #3 (Remarks to the Author):

This paper firstly shows that HC is simulated to be weakening under anthropogenic forcings using both CMIP5 and CESM simulations over an 850-2100 timespan. As pointed out by the paper, HC actually intensified over the last millennium using the signal-to-noise ratio approach. Therefore, this paper further investigates if the intensification is caused by internal variability or natural forcings and finds that internal variability alone cannot explain the intensification. This paper then used KE equation to break down the natural forcings and finds that latent heat and static stability are the two main contributors to the HC intensification. This paper further explains how static stability and latent heat affects the HC strength.

It's a good idea to examine Hadley circulation changes over a longer time period and breaks down the effects of the natural forcings. My suggestion is to accept this paper with minor revisions.

Major questions:

1. It's a good approach to compare between LIA and MCA since it can avoid the human-induced warming effects. However, if one wants to claim that human-induced warming weakens the HC but natural drivers strengthen it, one may also need to prove that during 1850-2100, Q_{lat} , static stability and eddy heat flux are still behaving the same way as between LIA and MCA.
2. Eddy heat flux seems minor comparing to Q_{lat} , but not too small comparing to static stability. Is it worth to check its behaviors from 1850 to 2100? How about the residual term?

Minor suggestions:

1. I was able to guess RHS means right hand side, but it would be better if the explanation is given before it is used.
2. Is it doable to make figure 4c's legend smaller?

Reviewer 1

General comments:

My main concern is the novelty of this work. First, Chemke and Polvani (2021), their Fig. 1a, and Chemke and Polvani (2019), their Figs. 1cd, have employed the same methodology and already shown that the magnitude of current NH HC weakening is unprecedented, though only by showing the time series from 1900 to 2100. The net radiative forcing has not changed much from year 850 to 1900 in comparison to the magnitude of the changes in 1900-2100 period, so the Hadley circulation is also not expected to change much too. The results of this study are therefore expected. Second, the same methodology of employing KE equation has been applied to multiple studies before and can be repeated on a bunch of other datasets too.

First, we would like to thank the reviewer for the careful reading and very useful comments. Second, we now further stress the novelty of our work in lines 40-42, 53-56, 231-235, 237-239, and 246-248. Specifically, here, for the first time, we (i) quantify how unprecedented is the human-induced weakening of the flow in recent and coming decades, relative to the naturally forced weakening over the last centuries, (ii) reveal that natural forcings acted to intensify the circulation over the past millennium; consequently, anthropogenic emissions have reversed a multi-century intensification of the flow, and (iii) elucidate the mechanism underlying the impacts of last millennium natural forcings on the flow.

Please note that previous work only examined the evolution of the HC over the industrial era, and thus could not address the question of how unprecedented is the human-induced weakening of the flow relative to the impacts of natural forcings in past centuries. By extending the time evolution of the HC to past centuries, we are able to unravel the unprecedented nature of anthropogenic emissions and reveal how natural forcings affected the flow. Furthermore, previous work did not quantify the unprecedented nature of the weakening of the flow, while here, this is done using detection analysis for the first time. Similarly, while previous studies used the KE equation to examine the HC changes in recent and coming decades, here we use it, for the first time, to elucidate the impacts of natural forcings on the flow; the novelty here is not in the method we employed (i.e., the KE equation) but in the emerged physical understanding it yields (changes in static stability and latent heating drive the naturally-forced strengthening of the flow). Finally, although the radiative forcings have changed less over the last millennium, relative to recent and coming decades, they were still able to give rise to a significant response in the long-term effect, and our finding regarding the multi-centennial naturally-forced strengthening of Ψ_{\max} over the last millennium proves this claim.

I have also several methodological concerns. First, the authors have not included CMIP6 simulations, which use a more advanced PMIP4 forcings instead of PMIP3. They argue in lines 234-236: “We here choose to analyse CMIP5 data rather than CMIP6 data, as only two CMIP6 models have available output for the analysis over the last millennium.” Second, they constrain their CMIP5 input data even more by using only those, which output meridional winds (lines 233-234). While I understand such choice as the authors employed stream-function to estimate the Hadley cell intensity, note that there exists a plethora of other metrics of Hadley cell intensity and these can be used instead of stream function to provide a more robust analysis of Hadley circulation changes.

Third, the authors opted to use RCP5-8.5 scenario as a proxy for “anthropogenic forcings”. I would here suggest using RCP4.5 or SSP2-4.5 if they include the CMIP6. These two scenarios are most aligned with current and projected anthropogenic forcings.

Following the reviewer’s comment, we now add CMIP6 simulations to our analysis and show that our results also hold under the PMIP4 forcing (lines 95-96, 274-280, and Extended Data Fig. 3). Additionally, we have repeated our analysis, employing a spatially averaged HC strength metric (lines 261-262 and Extended Data Fig. 11) and considering a more moderate warming scenario throughout the 21st century (lines 91-92, 271-274, and Extended Data Fig. 2).

Specifically, Fig. 1 below shows the evolution of the CMIP6 mean NH HC strength (using three available models) over the 850-2100 period, which exhibits similar behavior as Fig. 1 in the manuscript: (i) the human-induced weakening of the circulation in recent and coming decades is unprecedented compared to the HC-forced changes over the last millennium (Fig. 1a below), and (ii) over the last millennium the HC intensified at a rate of $1.6 \times 10^6 \text{ kg s}^{-1} \text{ yr}^{-1}$ in CMIP6 mean (Fig. 1b below). Thus, using CMIP6 models yields the same main conclusion, which is that the human-induced weakening of the HC reverses a multi-centennial forced strengthening of the circulation.

Please note that our aim is to examine the forced response of the HC to external forcings, which requires an ensemble of simulations to eliminate the internal variability in the system. The small ensemble size in CMIP6 over the last millennium thus prevents us from inferring the behavior of the forced response. Following the reviewer’s comment, we show this analysis in the supplementary to corroborate the findings from CMIP5 and CESM in the main text (Extended Data Fig. 3).

Second, unlike metrics for the HC width, the HC strength has almost entirely been examined using wind fields. For example, Pikovnik et al. (2022) analyzed 8 different metrics for the HC, all of which are based on wind data. Some metrics are based on the vertical wind rather than the meridional wind, but the availability of vertical wind model output over the last millennium is smaller than for the meridional winds. Even the availability of sea level pressure data, which was recently discovered by Chemke and Yuval (2023) to serve as a proxy for the HC strength, is smaller in CMIP5 and CMIP6, relative to the variability of the meridional wind over the last

millennium runs.

To ensure that our results are not metric dependent, we have repeated our analysis using a spatially averaged HC strength metric (lines 261-262 and Extended Data Fig. 11). Specifically, we follow Pikovnik et al. (2022) and average the meridional mass streamfunction between 10° and 25° and between 1,000 mb and 100 mb (Ψ_{avg} , Fig. 2 below). Overall, changing the HC strength metric did not change the results. Ψ_{avg} also shows that the human-induced HC weakening in recent and coming decades is unprecedented compared to the HC forced changes over the last millennium (panel a), and it also yields a HC strengthening over the last millennium at a rate of $0.8 \times 10^6 \text{ kgs}^{-1}\text{yr}^{-1}$ in CESM mean and at a rate of $0.6 \times 10^6 \text{ kgs}^{-1}\text{yr}^{-1}$ in CMIP5 mean (panel b). Also, note that we show in Extended Data Fig. 10 that the weakening in the circulation is evident throughout most of the HC. Altogether, this provides us confidence that our results do not depend on the metric used for the HC strength.

Finally, note that the use of the RCP8.5 scenario as a proxy for “anthropogenic forcings” does not change our results. Fig. 3 below shows the evolution of the HC under the RCP4.5 scenario. Even under a milder warming scenario, anthropogenic emission acts to weaken the circulation and thus also reverse a multi-century forced intensification of the flow. Second, the human-induced weakening is found to emerge from the forced last millennium changes during the 2010-2020 period. Over this period, the radiative forcings are very similar between RCP8.5 and RCP4.5. Thus, re-calculating the emergence of the HC weakening under the RCP4.5 yields similar results of emergence, around 2020, out of the last millennium forced NH HC changes (Fig. 4 below, lines 91-92, 271-274, and Extended Data Fig. 2).

Figure 1: **a**, Evolution of the NH HC strength (Ψ_{max}), relative to the 1810-1850 period, in CMIP6 mean. Shading shows s.d. across models. **b**, The 850-1849 Ψ_{max} trends in CMIP6. The black dot shows the mean trend, and the crosses show the individual models’ trends. The error bar shows the 95% confidence interval of the mean trend based on a Student’s t-distribution.

Figure 2: **a**, Evolution of the Ψ_{avg} (the averaged streamfunction between 10° and 25° and between 1,000 mb and 100 mb), relative to the 1810-1850 period, in CESM mean (red line) and in CMIP5 mean (black line). Shading shows s.d. across members/models. **b**, The 850-1849 Ψ_{avg} trends in CESM (red) and CMIP5 (black). The red and black dots show the CESM and CMIP5 mean trends, respectively, and the crosses show the individual members'/models' trends. Error bars show the 95% confidence interval of the mean trend based on a Student's t-distribution.

Figure 3: Evolution of the NH HC strength (Ψ_{max}), relative to the 1810-1850 period, in CMIP5 mean (between 2006-2100 under the RCP4.5 future scenario).

Figure 4: Signal-to-noise ratio analysis to the NH HC strength trend from 1970 and to each year plotted against the last year of trend in CMIP5 mean (between 2006-2100 under the RCP4.5 scenario). Shading shows the s.d. of signal-to-noise ratio values (Methods). The horizontal black line represents a signal-to-noise ratio value of -2.

I was surprised by the inaccuracy of the opening statement (“The Hadley Circulation (HC) is largely responsible for transferring energy (in the form of heat and moisture) from low to subtropical latitudes in both hemispheres.”) The Hadley circulation barely transports moisture poleward (“from low to subtropical latitudes”), as most of the moisture in the isolated convective towers rapidly condenses and falls out as precipitation in the Tropics. The lack of poleward moisture transport can be observed from time- and zonal-mean meridional transport of specific humidity ($v \partial q / \partial y$). The bulk of moisture transport in the Hadley circulation is equatorward in the lower branches of the Hadley cells. In relative terms, i.e. if the transport is normalised by the amount of humidity in a certain grid box ($v/q \partial q / \partial y$), the upper-tropospheric moisture transport actually becomes significant, but does not meaningfully shape the planetary climate due to low absolute amount of moisture in the upper-troposphere.

We agree, and following the reviewer’s comment, we revised the above lines to describe our intention more accurately (lines 20-21). The meridional propagation of air within the HC carries heat and moisture across different latitudes, which largely contributes to the distribution of precipitation in the tropics and subtropics. The total energy flux is poleward only when accounting for the geopotential energy flux.

I am also not fully convinced by the methodology of attributing NH HC strengthening to natural sources, described in lines 283-291. It would be helpful if the authors would provide some mathematical description of what they did. If I understand correctly, the authors did 4 forcing experiments with an ensemble. Lets denote the natural forcings f_1, f_2, f_3, f_4 . In each of the experiments, one of the forcing terms, say f_i , was allowed to evolve and the others were kept constant, $f_i = f_i(t)$ and $f_{j \neq i} = f_j(0)$, where $t = 0$ indicates the starting year 850. The sum of all forcings should be $f = f_1 + f_2 + f_3 + f_4 + f_a$, where f_a denotes anthropogenic

forcing. My first question is what happened to the anthropogenic forcings present in PMIP3? Was the term f_a allowed to evolve over time or was it kept fixed in all experiments (or was it even evolved in all experiments)? This essential part of methodology is missing making it very unclear. The error bars in Ext. Fig. 1c for the “Natural only” experiments are large. The relative contribution from each natural forcing should be shown separately including the anthropogenic forcing. How is the relative contribution defined – describe mathematically!

Following the reviewer’s comment, we revised the above lines and further clarified the methodology used in the attribution analysis for the NH HC strengthening over the last millennium (lines 104-110 and 329-343), and explicitly show the contribution from each external forcing agent (Extended Data Fig. 4). While external forcings over the last millennium were mostly of natural origins, some anthropogenic effects via land-use land-cover were also present. Thus, we conduct an attribution analysis widely used for quantifying the effects of external forcings on the climate system (Gillett et al., 2016). Specifically, for analyzing whether the forced HC strengthening over the last millennium can be attributed to natural forcings, we use 5 CESM single-forcing ensembles (which were carried out by Otto-Bliesner et al. (2016), and were used, for example, to assess the role of each forcing agent on the cooling of the last millennium). In each ensemble (of 3-5 members), a different natural/anthropogenic forcing agent (solar forcing, volcanic forcing, greenhouse gases forcing, orbital and land-use land-cover forcings) freely evolves throughout the 850-1849 period, while all other forcing agents are held fixed at 850 values (apart from the volcanic forcing that was kept off). Thus, the relative contribution of each forcing agent to the HC intensification is estimated as the mean HC trend in each single forcing ensemble (f_i). It is important to clarify that the CESM-LME model provides single-forcing ensembles, i.e., we didn’t run the experiments independently.

In addition, following the reviewer’s comment, we now explicitly show the contribution from each external forcing agent (Fig. 5 below and Extended Data Fig. 4). This also includes a comparison between the combined response to all anthropogenic forcings (land-use land-cover forcing) and the response to natural forcings (solar forcing, volcanic forcing, greenhouse gases, and orbital forcings). Note that the overall impact of natural forcings is estimated by aggregating the contribution from each natural forcing agent. The HC intensification over the last millennium is mostly due to the combined effect of natural forcings; these forcing agents are responsible for most of the radiative forcing over the last millennium. Although anthropogenic forcing also contributes to the intensification, it has a minor effect relative to the combined effect of natural forcings. Lastly, we note that different single-forcing ensembles hold a different number of members (the different and small numbers of members yielded in the large error bar). Thus, to estimate the internal variability similarly in all ensembles, we applied a bootstrap sampling procedure (of 10 iterations) to calculate the forced response (the mean across members) of a 1000-year HC strength trend by each forcing agent.

Figure 5: The contribution to Ψ_{\max} trend in CESM-LME mean between 850-1849 from natural forcings (Natural only) and anthropogenic forcings (Anthropogenic only, i.e., land-use land cover). Right to the dashed line is the decomposition of the contribution from natural forcings: greenhouse gases (GHG), orbital (ORBITAL), solar (SOLAR), and volcanic (VOLC). Error bars show the 95% confidence interval based on a Student's t-distribution.

Throughout the paper, the climate variability before the industrial age, i.e. in the past centuries, is exchanged for natural variability. For example, in lines 42-44, the authors wrote “Nevertheless, climate model simulations of the last millennium allow one to assess the response of the climate to natural forcings in recent centuries.” Similar lines of text can be found in the Methodology, e.g. “To estimate the internal variability in the system (the noise) in the detection of the naturally-forced HC strengthening over the 850-1849 period, . . .” Note that prior to industrial revolution, humans affected the global climate by land-use change (deforestation, agriculture) and the population boom (the population quintupled between year 800 and 1850), affecting the surface albedo and the biogeochemical cycle (Pongratz, 2008; Pongratz, 2009; Pongratz and Caldeira, 2012; Doughty, 2013). This is accounted for in PMIP3 and stated in the literature cited by the authors (Schmidt et al., 2011).

We thank the reviewer for this important point. We agree that it is inaccurate to state that the climate response is only due to natural forcings over the last millennium because there are multiple records of human influences throughout this period (Pongratz et al., 2008, 2009; Pongratz and Caldeira, 2012; Doughty, 2013). Thus, we revised our statements more accurately and referred to the relevant literature so it would be clear that over the last millennium, the climate response is not necessarily bound solely to natural forcings (lines 40-46, 49-56, 93-94, and 104-110). However, for analyzing whether the forced HC strengthening over the last millennium can be attributed to natural forcings, we use CESM-LME single-forcing ensembles (Otto-Bliesner et al. (2016), as discussed above). Specifically, the last millennium's forcing includes the contributions from solar forcing, volcanic forcing, greenhouse gases forcing, orbital and land-use land-cover forcings. The only forcing agent related to human influences is the land-use land-cover forcings, even though it includes natural influences throughout natural forest fires.

The rest of the forcing agents are entirely related to natural influences. Note that greenhouse gases variations might also be related to human influences, although the attribution of these relatively small changes before the industrial period is difficult (as noted in Schmidt et al. (2012)); hence, we still define them as natural forcing.

The attribution analysis results reveal that most of the HC intensification over the last millennium can be attributed to natural forcings (Fig. 5 above and Extended Data Fig. 4). Notably, the contribution of the anthropogenic forcing (land-use land-cover) is smaller relative to the summation of all the natural forcings' agents together. Although the anthropogenic forcing is not negligible relative to each natural forcings agent separately, it explains only a small portion of the HC intensification rate over the last millennium.

Lastly, Otto-Bliesner et al. (2016) conducted a similar attribution analysis using the CESM-LME single-forcing ensembles but for the NH annual surface temperature changes between the Medieval Climate Anomaly (MCA, 950-1250) and Little Ice Age (LIA, 1450-1850). Their results (cf. their Fig. 5) suggest that the warming of the surface between the two periods could predominantly be attributed to natural forcings (i.e., solar forcing, volcanic forcing, greenhouse gases, and orbital forcings). Together with our KE equation analysis, which links surface cooling and the HC intensification over the last millennium, it is shown that the HC strengthening over the last millennium is also attributed to variations in natural forcings.

In scope of that, the title “Anthropogenic forcings reverse a multi-century naturally-forced Hadley cell intensification” is inaccurate too: 0) it should be explicitly mentioned that the title is valid only for northern hemisphere Hadley cell; 1) the multi-century changes prior to industrial revolution were not entirely natural as might be implied from the title and 2) the title implies that the Hadley cell intensification in the last millennium is a fact, while no comparison with paleoclimate data is provided or reference to other studies. This is particularly strange as the authors (lines 236-239) justify the use of last millennium forcing by the availability of “vast array of proxy records (e.g., tree rings, ice cores, boreholes, etc.) over this period” which “places it as one of the ideal periods for studying the response of the climate system to natural forcings.”

Once again, we express our gratitude to the reviewer for this comment. First, acknowledging the importance of explicitly stating that our investigation focuses on the NH HC strength response, we have accordingly revised the manuscript's title. Second, according to the attribution analysis result discussed above, we show that the intensification of the circulation in past centuries is mostly due to natural forcings. Hence, anthropogenic forcings reversed a naturally forced intensification of the flow. Third, in the title, we mention that we examine the forced response of the HC to external forcings. The forced response of the system to external forcings (anthropogenic and/or natural) is derived by taking the mean across a single-model or multi-model ensemble, i.e., we here focus on a model result, not an observed result. Note that observed wind records are not available over the last millennium, and thus, comparison

with paleoclimate data is not possible for the HC strength response (as we note in lines 48-49). However, the available and rich archive of annually dated proxy records through the last millennium is used as a good basis for the last millennium forcing protocols defined by PMIP. The high-resolution and rich data provided by global climate models allow us to study short and long-term atmospheric circulation variability associated with the last millennium forcing, with a focus on natural forcings.

Some statements in the text should also be modified, for example the one in 31-33: “In spite of the biases found in reanalyses, which resulted in an artificial strengthening of the circulation in recent decades, the weakening of the circulation, as simulated by models, has been observed in recent decades and attributed to anthropogenic emissions.” While the source of reanalyses biases are both biased observations and biased models, the study of Chemke and Polvani (2019) did not provide a strong evidence of reanalyses biases, despite the claims. Their conclusion was based on a comparison between reanalyses precipitation and GPCP precipitation. Note that the GPCP precipitation data over the oceans are a product of satellite retrieval techniques, which in large part use the same raw data as the reanalyses. As such, the GPCP data over the oceans are prone to large uncertainties, as shown in Prakash et al. (2013), and later by Good et al. (2021, their Fig. 1). Taking GPCP observations as a truth for reanalyses verification, particularly over the oceans, is a very questionable approach.

We acknowledge the need to clarify the statements regarding the biases found in reanalyses. Thus, we modified the above lines and provided a more accurate perspective on recent HC strength changes, specifically, as simulated by reanalyses (lines 31-39). While, indeed, GPCP may be biased (as was shown in Prakash et al. (2013) and Good et al. (2021)), it is currently our best estimate for long-term trends of precipitation, and it shows different trend than in reanalyses, again likely due to latent heating biases in reanalyses. Regardless, in a recent study, Chemke and Yuval (2023), used measurements of sea level pressure to constrain the HC strength. They derived a relationship between the HC strength and sea level pressure from the momentum equations and used observations of sea level pressure as a proxy for the HC strength. They revealed that the observed sea level pressure proxy is consistent with models but not with reanalyses, showing a weakening of the flow in recent decades. This result further corroborates the findings of Chemke and Polvani (2019) that the intensification in reanalyses is likely artificial and that the models capture the correct sign of the trend.

There are also positives which should be mentioned. The main goal of this paper is interesting in my opinion, but the analysis should be extended, and possibly complemented with the use of paleoclimate data. The detection analysis is performed robustly, and the moisture budget analysis is very informative too.

We thank the reviewer's sincere comment. As mentioned above, we here show and quantify for the first time the unprecedented nature of the human-induced HC weakening, reveal that anthropogenic emissions reversed a multi-century intensification by natural forcings, and elucidate the physics underlying the naturally forced intensification. Following the reviewer's comments, we added analyses using CMIP6 data and with additional HC metric. We also show the attribution analysis result (to assess whether the forced simulated HC strengthening over the last millennium can be attributed to natural forcings) in a more coherent form. Finally, we stress that we can not complement our analyses using paleoclimate data because wind records from the last millennium are unavailable, and here, we focus on the forced response of the HC to anthropogenic vs. natural forcings.

Specific comments:

21: too simplified: the global HC consists of many regional cells with distinct characteristics, which govern the zonal and meridional distribution of precipitation

We agree, and following the reviewer's comment, we have revised the above line to exclude the term "latitudinal" while maintaining clarity in describing the impact of vertical motions within the HC on the distribution of precipitation over the tropical and subtropical latitudes (lines 21-22).

23: near the equator (ITCZ specifically)

Done (line 23).

30-33: The biases in the reanalyses that occur in the data assimilation procedure due to biased (satellite) observations are undisputed. These can be only verified by comparing reanalysis products to trustworthy conventional observations, e.g. Simmons (2022). The biases also affect the intensity of Hadley circulation, among others. However, note that some studies have suggested that these biases are complemented by the natural long-term atmospheric variability, namely Atlantic Multidecadal Oscillation (AMO), which could also temporarily oppose the ongoing weakening of the Hadley circulation (Zaplotnik et al., 2022). The hint to the link between the AMO and HC has been given by Baines et al., 2007, and others. Presenting the recent Hadley circulation strengthening in the reanalyses as a sole consequence of biases is therefore rather inaccurate.

Following the reviewer's comment, we have modified the above lines (lines 31-39). Please note that the artificial trends in the HC strength in reanalysis, found in Chemke and Polvani (2019) using precipitation observations, were recently corroborated using measurements of sea level pressure. Chemke and Yuval (2023) derived a relationship between the HC strength and sea level pressure and used measurements of the latter to constrain the recent HC changes. They found that reanalyses show the opposite evolution to the observed one, while models adequately capture the observed trend.

In addition, recent work (Zaplotnik et al., 2022) showed that in ERA5, the HC exhibits multi-decadal changes in recent decades: it has strengthened up to the early 00's and slightly weakened thereafter. These multi-decadal variations were argued to be related to the AMO (Baines and Folland, 2007), but such a link (which was based on correlation) was only found in one reanalysis (ERA5); all other reanalyses do not support this conclusion, as discussed in Zaplotnik et al. (2022) (cf. their Fig. 8b). Furthermore, since the circulation was argued to be driven by the changes in latent heating, Zaplotnik et al. (2022) first compared changes in precipitation, as a proxy for latent heating in ERA5 and GPCP. As found in Chemke and Polvani (2019), over the last several decades, precipitation changes in reanalysis were inconsistent with observations (when including the intensification in reanalyses). Post-2000, when the circulation slightly weakened, changes in precipitation were found to be more consistent with observations

(Simmons, 2022); this weakening is consistent with the overall weakening in models and in the observed sea level pressure analysis by Chemke and Yuval (2023), and we thus do not see any contradiction with recent work. We hope, therefore, that this indicates that the new generation of reanalyses is starting to capture the correct changes in the circulation.

44-45: What does “an unprecedented increase in the magnitude of the global pattern of surface winds over the 20th century” mean? In particular, I am puzzled by the wording of “the magnitude of the global pattern”. How do you define a magnitude of a pattern? This part needs to be rewritten in a less dubious way.

In the above lines, we refer to the paper by José Roldán-Gómez et al. (2020), who conducted an EOF analysis of the longitude-latitude pattern of surface winds, and found an increase in the magnitude of the first EOF. Since the first EOF mostly represents the poleward displacement of the winds, we follow the reviewer’s comment and revised the above lines (lines 52-53), indicating that previous work found that the poleward shift of surface zonal winds is unprecedented over the 20th century, relative to past centuries.

38, 43 and 47: as stated in the Major comments’ section, the forcings in the last millennium before 1850 were not only natural.

We modified the relevant statements to explicitly acknowledge that the external forcings over the pre-industrial millennium (before 1850) were not solely natural, considering the potential influence of anthropogenic factors during that period (lines 40- 46, and 49-56).

221-223: the increase in intensity is very inhomogeneous over the NH HC in Ext. Fig. 4b for CMIP5 mean. Note that precisely for this reason, Nguyen et al. (2013), Pikovnik et al. (2022) and others have derived metrics of average HC intensity by employing either vertical or spatial averaging.

Following the reviewer’s comment, we have repeated our analysis using a spatially averaged HC strength metric (lines 261-262 and Extended Data Fig. 11) and show that our results are independent of the HC metric being examined. Please note, throughout most of the HC, in both CESM and CMIP5, an intensification is evident over the last millennium. Specifically, as discussed above, we follow Pikovnik et al. (2022) and Nguyen et al. (2013) and average the meridional mass streamfunction between 10° and 25° and between 1,000 mb and 100 mb (Ψ_{avg}). Fig. 2 above shows that an averaged metric of the HC also shows that the human-induced HC weakening in recent and coming decades is unprecedented compared to the HC forced changes over the last millennium (panel a), and it also yields an HC strengthening over the last millennium at a rate of $0.8 \times 10^6 \text{ kgs}^{-1}\text{yr}^{-1}$ in CESM mean and at a rate of $0.6 \times 10^6 \text{ kgs}^{-1}\text{yr}^{-1}$ in CMIP5 mean (panel b).

219: gravity acceleration = gravitational acceleration + centrifugal force

Thank you; done (line 257).

248-249: the spread across the members is only a proxy for current-state internal climate variability. The internal climate variability can itself attain a new state after climate undergoes forced response, bringing it into another state.

Indeed, similar to the forced response in the system, the internal climate variability may change with time due to changes in external forcings, for example. Both of these changes are accounted for in model ensembles. Thus, our estimate of the forced response (and internal variability) is indeed time-dependent.

254: “850-control run”. Is that control run initiated in year 850?

As described in Otto-Bliesner et al. (2016), the 850-control run is a control run for 1,356 years forced with a constant 850 forcings (branched from the 1850 control simulation). This is now mentioned in lines 299-300.

283: NH HC

Done (lines 329-330). We also further stressed that our findings are regarding only the NH HC (see the manuscript’s title and lines 56-57).

Reviewer 2

This study newly presents a Hadley cell strengthening in the last millennium (850 ~ 1850 year), which is driven by natural forcing-induced cooling. The authors convincingly show the increasing trend of the Hadley cell by employing CMIP and large-ensemble experiments. Additionally, their application of theoretical approaches provides valuable understanding of the various contributions to this trend. The scientific findings presented here are of considerable interest not only to the tropical climate community but also to researchers in the historical science field.

However, I think the current manuscript is not substantiative enough to warrant publication in Nature communications in the current form. The main concern I have is the need for a more comprehensive explanation of Northern Hemisphere (NH) Hadley cell changes. As such, I have included several comments below, based on my reading of the manuscript. Please review the provided feedback for further consideration.

We thank the reviewer for the careful reading and very useful comments.

Major comments

1. As suggested by the authors in L117-118, it may not be surprising that global cooling induces the strengthened Hadley circulation in the linear thinking of substantial research on the weakening of the Hadley cell under global warming (e.g., Lau and Kim 2015; Lu et al. 2008). Also, as suggested in L150-154, the tropical cooling (following the wet adiabat) reduces the static stability, which is exactly the opposite of what happens in global warming scenarios (Figure 3b,d in Chemke and Polvani 2021).

It is clear that the natural forcing of the last millennium regulates the Hadley cell through the Q, which is the biggest difference to global warming. Although the authors show that the Q response is mainly driven by mean circulation changes, the suggested feedback is too vague to understand the Hadley cell response.

For me, it looks like there is a clear southward shift of the ITCZ, especially in the CMIP (Extended Data Figure 4). So it is very plausible that the dynamical feedback is related to the ITCZ. Then, the Hadley cell response over the last millennium could not be directly compared with that of global warming, considering its relatively symmetric changes (e.g., Figure 1 in Lau and Kim 2015). As previous studies already show that the interhemispheric energy difference could induce the ITCZ response (e.g., Kang 2020), I think further analysis to converge the authors' theory and previous findings on the ITCZ response to the thermal forcing is needed and really helpful to understand the Hadley cell dynamics. Or, at least, detail explanation and discussion on the dynamic feedback is necessarily included.

We thank the reviewer for this comment and for bringing the above papers to our attention, and we now discuss the above papers (Lau and Kim, 2015; Lu et al., 2008; Kang, 2020) and the potential impact of the ITCZ shift in the strengthening of the HC over the last millennium

(lines 221-226 and Extended Data Fig. 8) and further elucidate the dynamic feedback around the changes in latent heating (lines 201-206). As discussed in Kang (2020), uneven heating between the hemispheres could be compensated by cross-equatorial energy transport by the HC, which shifts the ITCZ. As the reviewer pointed out, according to Extended Data Fig. 10, it may be possible to distinguish an ITCZ southward shift over the last millennium. It is thus conceivable that a shifted ITCZ might alter the meridional gradient of precipitation (or latent heating), thus affecting the HC strength. In addition, changes in the HC strength might be expected due to the transport of more energy from one hemisphere to another. However, further investigation of the ITCZ location (ϕ_{ITCZ} , defined as the latitude where the 500 mb meridional mass streamfunction, Ψ , is zero) evolution over 850-2100 reveals that there was no clear ITCZ shifting trend over the last millennium (shown in Fig. 6 below). This finding, together with the hemispheric symmetric response of the troposphere's temperature between the MCA and LIA (Fig. 4a in the manuscript and Extended Data Fig. 7a), cast doubt on the above mechanism as being responsible for the strengthening of the NH HC over the last millennium. Thus, we still find it appropriate to directly compare the NH HC strengthening under global cooling over the last millennium with the NH HC weakening under global warming in recent and future decades. Moreover, previous studies (Chemke and Polvani, 2019, 2021) showed using the KE equation that the dominant component that contributes to the recent and projected HC weakening is static stability. Using a similar analysis, we also find that this term contributes to the intensification of the HC over the last millennium. Therefore, the similarity between the controlling mechanisms between the two periods supports our comprehension that the HC strengthening over the last millennium might be linked to the global cooling between the MCA and LIA.

Furthermore, following the reviewer's comment, we revised the discussion on how the NH HC might have acted as positive feedback over the last millennium (lines 201-206). As mentioned above, we follow previous works and investigate the Ψ_{max} response using the KE equation. Accordingly, we reveal that the primary mechanisms that contributed the most to the Ψ_{max} strengthening over the last millennium are the meridional gradient of latent heating and static stability. To explain the changes in latent heating, we use the vertically integrated moisture budget equation to analyze the changes in surface precipitation (since the net latent heating in an atmospheric column equals surface precipitation). We find that changes in the mean circulation increased latent heat release at low latitudes, resulting in a larger meridional gradient (Fig. 4c in the manuscript), which further strengthened Ψ_{max} according to the KE equation. Considering the collective insights from the above findings, we suggest that over the last millennium, the intensification of the NH HC might have acted as positive feedback; the Ψ_{max} intensification (which may be triggered by a decrease in static stability) increased latent heat release over the ascending branch of the HC, resulting in a larger meridional gradient of latent heating, which further strengthened Ψ_{max} .

Figure 6: **a**, Evolution of the ϕ_{ITCZ} (defined as the latitude where the 500 mb meridional mass streamfunction is zero), relative to the 1810-1850 period, in CESM mean (red line) and in CMIP5 mean (black line). Shading shows s.d. across members/models. **b**, The 850-1849 ϕ_{ITCZ} trends in CESM (red) and CMIP5 (black). The red and black dots show the CESM and CMIP5 mean trends, respectively, and the crosses show the individual members/models. Error bars show the 95% confidence interval of the mean trend based on a Student's t-distribution.

2. In line with comment 1, it looks like the authors are using the NH HC changes and the HC changes without any clarification, which makes me quite confused. I think that because the ITCZ shift can induce asymmetric changes between the two Hadley cells, NH HC might not be enough to represent the overall Hadley cell changes. However, for example, L174-177 discuss the ascending branch of the HC related to the two cells, where they only consider NH HC ascending branch. Also, the discussion generally talks about the HC changes rather than NH HC changes, although it cannot simply explain the SH HC changes. Please provide solid reasons for considering NH HC as representative of the general tropical circulation. Without this, clarification of the overall statement would be helpful.

We thank the reviewer for this important point. Accordingly, we revised the manuscript so it will be clear that all of our findings are regarding only the NH HC (see the title and in lines 40-42, 53-56, 56-57, and 60), and we now discuss the SH HC evolution between 850-2100 (in lines 242-245 and Extended Data Fig. 9). We here focus on the NH HC since anthropogenic emissions are found to considerably change the strength of the flow only in the NH (Chemke and Polvani, 2021). Thus, since our aim is to quantify the unprecedented nature of human-induced circulation changes, we focus on the NH (where there is a human-induced signal). Additionally, in contrast to the result from the NH HC strength analysis, examining the SH HC strength evolution over the last millennium (Fig. 7a below) reveals that there is no significant multi-century trend or even an agreement between the different models/members on the sign of the trend (Fig. 7b below). We also would like to stress again that there is no clear shift in the ITCZ over the last millennium (Fig 6 above).

Figure 7: **a**, Evolution of the SH HC strength (Ψ_{\min} , defined as the minimum value, at 500 mb, of the meridional mass streamfunction), relative to the 1810-1850 period, in CESM mean (red line) and in CMIP5 mean (black line). Shading shows s.d. across members/models. **b**, The 850-1849 Ψ_{\min} trends in CESM (red) and CMIP5 (black). The red and black dots show the CESM and CMIP5 mean trends, respectively, and the crosses the individual members/models. Error bars show the 95% confidence interval of the mean trend based on a Student's t-distribution.

Minor comments

1. L79: it looks like a typo 'the early 00s'

No, "the early 00s" is not a typo but a shorthand way that refers to the early 2000s, often expressed as the first decades of the 21st century.

2. L113-117: Considering the large intermodel spread in the time of emergence, is it appropriate to compare just two periods rather than employing linear trend over the last millennium?

Following the reviewer's comment, we highlighted the rationale behind our focus on the change in Ψ_{\max} between the MCA and the LIA (lines 137-140 and Extended Data Fig. 5). Our decision aligns with the common approach to discussing climate change over the last millennium; given the numerous pieces of evidence pointing to changes in surface temperature between the MCA and LIA, primarily attributed to natural forcings, we deemed it pertinent to investigate the shifts between these two climatic phases. Moreover, the spread between the models/members exhibited a similar pattern for both the trend and for $\Delta\Psi_{\max}$ (compare Fig. 1b in the manuscript to Fig. 8 below). This similarity supports our choice to establish our assumption on $\Delta\Psi_{\max}$ rather than the linear trend, given their general equivalence, and allows us to discuss the changes in the context of the extensive research comparing the two periods.

Figure 8: The 850-1849 $\Delta\Psi_{\max}$ (the difference between the LIA and MCA) in CESM (red) and CMIP5 (black). The red and black dots show the CESM and CMIP5 mean, respectively, and the crosses show the individual members/models. Error bars show the 95% confidence interval of the mean difference based on a Student's t-distribution.

3. L157: Definition of NH HC ascending branch should be noted in the main manuscript.
Done (lines 182-185).

4. Figure 3a: including $y=x$ line would be helpful.

Following the reviewer's comment, we now include $y=x$ line in Fig. 3a in the manuscript and discuss it in lines 157-158. The addition of the 1:1 line reveals that the KE equation slightly overestimates the change in the HC strength (Fig. 9 below), which can be explained by the fact that the KE equation is derived by combining thermodynamic and momentum quasi-geostrophic approximations. Yet, it adequately captures the evolution of the HC strength over the last millennium.

Figure 9: Changes in the NH HC strength between the LIA and the MCA in CESM as inferred from the KE equation ($\Delta\Psi_{\max}^{\text{KE}}$) as a function of $\Delta\Psi_{\max}$; their correlation appears in the upper left corner. The blue line represents the $y = x$ line.

5. Figure 4a: I do not see any reason why the Hadley response should not be shown in this figure.

We thank the reviewer for this suggestion. However, Fig.4 in the manuscript aims to elucidate the mechanism behind the HC intensification. Adding the spatial changes in the flow would only divert the reader from the main take-home message of this figure. We thus prefer to show the spatial changes in the HC strength in the extended data (Extended Data Fig 10).

Reviewer 3

This paper firstly shows that HC is simulated to be weakening under anthropogenic forcings using both CMIP5 and CESM simulations over an 850-2100 timespan. As pointed out by the paper, HC actually intensified over the last millennium using the signal-to-noise ratio approach. Therefore, this paper further investigates if the intensification is caused by internal variability or natural forcings and finds that internal variability alone cannot explain the intensification. This paper then used KE equation to break down the natural forcings and finds that latent heat and static stability are the two main contributors to the HC intensification. This paper further explains how static stability and latent heat affects the HC strength.

It's a good idea to examine Hadley circulation changes over a longer time period and breaks down the effects of the natural forcings. My suggestion is to accept this paper with minor revisions.

We thank the reviewer for the careful reading and very useful comments.

Major questions:

1. It's a good approach to compare between LIA and MCA since it can avoid the human-induced warming effects. However, if one wants to claim that human-induced warming weakens the HC but natural drivers strengthen it, one may also need to prove that during 1850-2100, Qlat, static stability and eddy heat flux are still behaving the same way as between LIA and MCA.

We thank the reviewer for this comment, which is now discussed in lines 163-165, 175-177, and 207-220. First, please note that to show that natural forcings strengthened the circulation over the last millennium while anthropogenic emissions weakened it in recent and coming decades, we conduct detection-attribution analyses that allow us to quantify the impacts of the different forcing agents on the flow. This attribution is independent of the mechanism underlying the changes in the strength of the circulation. In other words, not necessarily natural and anthropogenic forcing agents modulate the flow via the same processes. Yet, Chemke and Polvani (2021) employed the KE equation to elucidate the physical mechanisms responsible for the HC weakening in response to anthropogenic forcings. Their main conclusion was that the HC weakening results from an increase in static stability but with a considerable cancellation by an increase in the meridional gradient of latent heating.

Interestingly, using the KE equation, we find an opposite impact of static stability but a similar tendency of latent heating on the HC strength over the last millennium, and those terms contribute the most to the intensification of the HC over that period. The opposite impacts of static stability to weaken the flow in coming decades but strengthen it over past centuries likely stems from the tendency of natural forcings to cool the surface and thus follow the moist adiabatic lapse rate to destabilize the troposphere (cf. Fig. 4a in the main text) and strengthen the

flow, while human-induced surface warming will act to destabilize the troposphere and weaken the flow (Extended Data Fig. 7b).

Second, the latent heating contribution, on the other hand, is to intensify the HC in both the last millennium and by the end of this century, despite the opposite temperature changes. We further reveal, using the moisture budget equation, that dynamical processes are responsible for the increase in the meridional gradient of latent heating between the MCA and LIA (Fig 4c in the manuscript), whereas the explanation for the increase in latent heating gradient over recent and future decades is due to thermodynamic arguments (i.e., the 'wet gets wetter, dry gets drier' mechanism by Held and Soden (2006)).

Finally, in both cases, the other components in the KE equation (e.g., eddy heat flux) have minor and even opposite contributions to the total response of the circulation. Therefore, the comparison of our findings on the mechanisms responsible for HC strengthening over the last millennium to the results of Chemke and Polvani (2021) on the mechanisms responsible for anthropogenically-forced HC weakening supports our comprehension that the HC strengthening over the last millennium is linked to cooling between the MCA and LIA, which was argued to stem from changes in natural forcings; changes in global surface temperature, induced by external forcings, drive changes in low latitudes circulation mainly through changes in static stability and meridional gradient of latent heating.

2. Eddy heat flux seems minor comparing to Q_{lat} , but not too small comparing to static stability. Is it worth to check its behaviors from 1850 to 2100? How about the residual term?

Again, we thank the reviewer for the thorough review. By exploiting the KE equation, we reveal that the only terms that contribute to the intensification of the HC over the last millennium are the meridional gradient of latent heating (Q_{lat}) and static stability (S^2) (Fig. 3b in the manuscript). Our choice not to elaborate on the other KE equation components' contributions (Q_{rad} , $v'T'$, $u'v'$, X , and the residual) stems from their smaller and opposite effect relative to Q_{lat} and S^2 , i.e., those terms acted to weaken the circulation (Fig. 3b in the manuscript). Please note that the contribution of these terms is half the contribution from static stability. Thus, we believe that further analyzing the role of eddy heat fluxes, which hold mitigating minor effects, would only divert the reader from the main take-home message of this section, i.e., providing a physical explanation for the intensification of the flow.

Moreover, previous studies (Chemke and Polvani, 2021, 2019), which investigate the responsible physical mechanisms for the human-induced HC weakening via the KE equation, found a similar behavior of $v'T'$ and *res* terms (i.e., relatively small contributions to the total weakening of the HC). In addition, as in the HC response over the last millennium, static stability and meridional gradient of latent heating are the dominant components in the HC response in recent and future decades (now discussed in lines 163-165). The similarity between the KE equation results for the last millennium and the 20th – 21st periods provides us confidence in our findings regarding the source of the HC strengthening in response to natural forcings over

the last millennium.

Minor suggestions:

1. I was able to guess RHS means right hand side, but it would be better if the explanation is given before it is used.

Done (lines 146-147).

2. Is it doable to make figure 4c's legend smaller?

We thank the reviewer for the suggestion. We suppose that resizing the legend could make it difficult to read. We are aware that in its current size, the legend hides some of the curves, but the main interest in the figure is still visible, which is the area of the NH HC ascending branch.

References

- M. Pikovnik, Ž. Zaplotnik, L. Boljka, and N. Žagar. Metrics of the Hadley Circulation Strength and Associated Circulation Trends. *Weather Clim. Dynam*, 3(2):625–644, 2022.
- R. Chemke and J. Yuval. Human-Induced Weakening of the Northern Hemisphere Tropical Circulation. *Nature*, 617(7961):529–532, 2023.
- N. P. Gillett et al. The Detection and Attribution Model Intercomparison Project (DAMIP v1.0) Contribution to CMIP6. *Geoscientific Model Development*, 9(10):3685–3697, 2016.
- B. L. Otto-Bliesner et al. Climate Variability and Change since 850 CE: An Ensemble Approach with the Community Earth System Model. *Bull. Am. Meteor. Soc.*, 97(5):735–754, 2016.
- J. Pongratz, C. Reick, T. Raddatz, and M. Claussen. A Reconstruction of Global Agricultural Areas and Land Cover for the Last Millennium. *Glob. Biogeochem. Cycl.*, 22(3):GB3018, 2008.
- J. Pongratz, C. H. Reick, T. Raddatz, and M. Claussen. Effects of Anthropogenic Land Cover Change on the Carbon Cycle of the Last Millennium. *Glob. Biogeochem. Cycl.*, 23(4):GB4001, 2009.
- J. Pongratz and K. Caldeira. Attribution of Atmospheric CO₂ and Temperature Increases to Regions: Importance of Preindustrial Land Use Change. *Env. Res. Lett.*, 7(3):034001, 2012.
- C. E. Doughty. Preindustrial Human Impacts on Global and Regional Environment. *Ann. Rev. Env. Res.*, 38(1):503–527, 2013.
- G. A. Schmidt et al. Climate Forcing Reconstructions for use in PMIP Simulations of the Last Millennium (v1.1). *Geosci. Model Dev.*, 5(1):185–191, 2012.
- S. Prakash, C. Mahesh, and Gairola R. M. Comparison of TRMM Multi-satellite Precipitation Analysis (TMPA)-3B43 Version 6 and 7 Products with Rain Gauge Data from Ocean Buoys. *Remote Sens. Lett.*, 4(7):677–685, 2013.
- P. Good et al. High Sensitivity of Tropical Precipitation to Local Sea Surface Temperature. *Nature*, 589(7842):408–414, 2021.
- R. Chemke and L. M. Polvani. Opposite Tropical Circulation Trends in Climate Models and in Reanalyses. *Nat. Geosci.*, 12:528–532, 2019.
- Ž. Zaplotnik, M. Pikovnik, and L. Boljka. Recent Hadley Circulation Strengthening: a Trend or Multidecadal Variability? *J. Clim.*, 35(13):4157–4176, 2022.

- P. G. Baines and C. K. Folland. Evidence for a Rapid Global Climate Shift across the Late 1960s. *J. Clim.*, 20(12):2721, 2007.
- A. J. Simmons. Trends in the Tropospheric General Circulation from 1979 to 2022. *Weather Clim. Dynam.*, 3(3):777–809, 2022.
- P. José Roldán-Gómez, J. Fidel González-Rouco, C. Melo-Aguilar, and J. E. Smerdon. Dynamical and Hydrological Changes in Climate Simulations of the Last Millennium. *Clim. Past*, 16(4):1285–1307, 2020.
- H. Nguyen, A. Evans, C. Lucas, I. Smith, and B. Timbal. The Hadley Circulation in Reanalyses: Climatology, Variability, and Change. *J. Clim.*, 26:3357–3376, 2013.
- W. K. M. Lau and K. M. Kim. Robust Hadley Circulation Changes and Increasing Global Dryness due to CO₂ Warming from CMIP5 Model Projections. *Proc. Natl. Acad. Sci. U.S.A.*, 112(12):3630–3635, 2015.
- J. Lu, G. Chen, and D. M. W. Frierson. Response of the Zonal Mean Atmospheric Circulation to El Niño versus Global Warming. *J. Clim.*, 21(22):5835, 2008.
- S. M. Kang. Extratropical Influence on the Tropical Rainfall Distribution. *Curr. Clim. Change Rep.*, 6:24–36, 2020.
- R. Chemke and L. M. Polvani. Elucidating the Mechanisms Responsible for Hadley Cell Weakening Under 4 × CO₂ Forcing. *Geophys. Res. Lett.*, 48(3):e90348, 2021.
- I. M. Held and B. J. Soden. Robust Responses of the Hydrological Cycle to Global Warming. *J. Clim.*, 19:5686–5699, 2006.

Reviewers' comments:

Reviewer #1 (Remarks to the Author):

General comments:

The revised version of the paper is improved in comparison to the original version. An extra effort has been performed to additionally analyse three CMIP6 runs and to add realistic (RCP4.5 and SSP2-4.5) emission scenarios.

While I think the results are based on an extensive analysis that in general merits a publication in Nature Communications, many claims (even in the Abstract and in the Title) are not unambiguously backed by the results. Some statistical analyses in the paper are inadequate and should be revised. The sensitivity of the trends to the applied time interval should be performed. Therefore, I would suggest a major revision of the paper.

Major comments:

1. Abstract, lines 12-13: The authors state “natural forcings acted to intensify the circulation”. I fail to fully see how this claim is backed by the results looking at Figures 1a and Extended Data Figure 11a. For example, looking at Figure 1a, the trendline difference between 850 and 1849 seems:
 - a. Smaller than the standard-deviation of the ensemble-mean time series
 - b. Much smaller than the ensemble spread.

Saying this, I am yet not fully convinced by the statistical approach applied in this study to test the significance of the trend in 850-1849. Instead of evaluating the trend for each model run and then performing Student-t test for the mean of the trends (as the Authors did in Figure 1b and Extended Data Figure 11b), one would have to test significance of the trend using Mann-Kendall test. The Mann-Kendall statistical test for trend is used to assess whether a set of data values is increasing over time or decreasing over time, and whether the trend in either direction is statistically significant. One should therefore:

- a. Apply Mann-Kendall test to every model run (is each of the trends significant between 850 and 1849? I suspect they are not because the internal variability signal has larger magnitude)
- b. Apply Mann-Kendall test to estimate the significance of the trend of the ensemble mean of the runs (in lines 72-73, the Authors even say, that they focus on the ensemble mean)

Note that the significance of the trend in Mann-Kendall test depends mostly on the magnitude of the trend in comparison to the standard deviation of the time series, which implicitly evaluates whether and to what extent the evaluated trend is dependent on the applied time interval. For example, would the trends be the same if you apply the starting interval at year 950 instead of 850? What about changing the end interval to 1750 instead of 1850. Would the conclusions stay the same? Are the results robust to the alteration of the start/end year or both? This would also have to be tested with respect to the analysis described in lines 111-117, and the analysis should be performed for !"# too. See for example Chung et al., 2019, their Fig. 2.

Considering that, it remains questionable whether the title is correct (“reverse a multi-century naturally forced intensification”). It might well be a stagnation. It also remains questionable whether the language in the remaining of the Abstract is correct. The text

should therefore be modified according to the outcome of the Mann-Kendall statistical test and sensitivity tests of the applied interval for trend evaluation. For an independent review, the time series used to generate Figures 1a and Extended Data Figure 11a should also be appended in the revised version. To me, besides $\%&$ for CESM, the trends do not seem significant. They seem susceptible to the applied interval.

Chung, ES., Timmermann, A., Soden, B.J. et al. Reconciling opposing Walker circulation trends in observations and model projections. *Nat. Clim. Chang.* 9, 405–412 (2019).

<https://doi.org/10.1038/s41558-019-0446-4>

2. Abstract lines 15-16. In any case, the effect of natural forcings during the 850-1850 does not seem to be large enough to vastly alter the Hadley cell strength. The language in the Abstract (“the large effect of natural forcings on the Hadley circulation”) should thus be more modest, e.g. “the amplifying effect of natural forcings on the Hadley circulation”. The same applies to Discussion section in lines 247-248.
3. The title. The results on NH Hadley cell intensification are not backed by observed data, e.g. paleoclimate data revealing the surface ocean currents in the tropical/subtropical oceans. Therefore, the title should unequivocally suggest that the outcome of the study is based on climate model simulations and is therefore not an undisputable fact, despite using reconstructions of external forcings. This has already been suggested in the previous review. I would suggest sth. like: “Anthropogenic forcings reverse a [simulated] multi-century naturally-forced Northern Hemisphere Hadley cell intensification.” If the authors want to state it as a fact, they should perform a more extensive study using paleoclimate data.
4. Lines 201-206: The arguments presented in this paper are not sufficient to call it a positive feedback loop. First, the KE equation is diagnostic, and it only describes that the diabatic heating meridional gradient mostly changes proportionally with the Laplacian of the stream-function. KE equation does not describe any lead-lag relationship between the two, but obviously the two change in accordance, and this is captured by the KE equation. Second, the change in the mean-circulation ($\# \Delta$) is also the change of the circulation itself. If you integrate Δ vertically and $\#$ was constant vertically, you would get Δ . However, $\#$ decreases approximately exponentially with height, so what you get is the contribution to Δ from the lower HC branch, which brings the fuel for condensation, which in turn helps maintaining the meridional gradient in diabatic heating. However, again, the moisture budget does not include any lead-lag mechanism. All this does not mean that the positive feedback loop is not actually present, it only means that the evidence presented in the manuscript is not sufficient to call it a feedback loop. Adjustment experiments using a fully coupled climate model would be needed to elucidate the mechanism.

Other comments:

1. Line 35: I would not say “as opposed to wind observations”, try to reformulate it. The trend in the poleward winds in the upper branch of the Hadley circulation is too small to be unequivocally

distinguished from the analysis-minus-background standard deviation. On the other hand, the trends in the zonal easterlies are significant and larger than the observation-sampling noise.

2. It would be good to see CMIP6 with *avg* as it is done with *ma&* in Extended Data Figure 3.
3. Lines 58-61: The authors first state, that the changes in volcanic forcings are not considered in climate model projections. What do the Authors then mean by “would allow one to better constrain the future climate changes”.
First, do you mean projections of future climate change?
Second, how could the projections be constrained? For example we do not know, when some volcano is going to erupt, how much material will it depose into the atmosphere, what will be the emission altitude, what would be the ratio between warming constituents (e.g. stratospheric water vapour) and cooling constituents (e.g. sulfate aerosols). Perhaps, the Authors had some other natural forcing in mind, which could actually better constrain the projections. I suggest some reformulation of the text.
4. 148-149 and Figure 3: the methodology how the authors obtained the contribution of Q_{lat} and Q_{rad} from Q , obtained by KE decomposition, is not clear. Were Q_{rad} and Q_{lat} available as model outputs? Was column-integrated Q_{lat} estimated from precipitation? Describe the methodology.
5. “gravity” instead of “gravity acceleration” would suffice. I used “gravity acceleration = gravitational acceleration + centrifugal force” only to demonstrate the difference between the terms gravitational acceleration and gravity.
6. Line 192: I would recommend using the same form of equation as in Eq.6. While the Authors rightfully say the equation is the vertically integrated moisture budget equation, the ∇ , went missing here neither is mentioned.
7. 223-225: NH and SH exhibit similar changes in temperature only in the tropics and subtropics (which matter mostly for the HC). Please reformulate accordingly .
8. Lines 261-262: I am not sure “similar” is the right expression. The emergence of the trend out of the noise is not similar using the average cell strength *avg* as it appears later than as in the case of *ma&* (2010-2020), right?
9. Lines 261-262: In response to Reviewers’ comments, the Authors state: “Specifically, we follow Pikovnik et al. (2022) and average the meridional mass streamfunction between 10° and 25° and between 1,000 mb and 100 mb (Ψ_{avg} , Fig.2 below).”

This is not the definition of the average Hadley cell strength defined in Pikovnik et al.

(2022), see their Eq. 3. Please, revise accordingly.

10. Lines 288-301: which variables did the Authors obtain? As far as I understand from Fig 3b, Q_{lat} and Q_{rad} were available?
11. 362: define how Q was computed – as a thermodynamic residual?
12. 364: Running or static monthly-mean?
13. 385-386: Same question: are the eddies computed as deviation from a static or running annual-mean zonal-mean?

Reviewer #3 (Remarks to the Author):

This revised paper focused on the impact of natural forcings on Northern Hemisphere (NH) Hadley Cell (HC) over the last millennium. This time, CMIP6 simulations were also analyzed. This version firstly stated that, over the last millennium, it is the natural forcings that strengthen the NH HC instead of internal variability and human-induced forcings. To find the source of the natural variability, this study again used KE equation to decompose the HC strength to different natural forcings and found that static stability and latent heat are the two major factors. My suggestion is to accept the paper with major revisions. The reasons are as follows:

This is a good idea to examine the ITCZ responses per other reviewers' request. However, I feel the section "Linking the Hadley cell strengthening to the cooling over the last millennium" needs a more comprehensive explanation. Currently, the static stability's role is well explained but not the latent heat part.

Firstly, for the location changes of the ITCZ, can you explain clearer on how the HC intensity would affect the location/shift of the ITCZ? I am also wondering if it is the location of the ITCZ or the width of the ITCZ is more related to the intensity of the HC.

I can see the latent heat intensifies the NH HC but how it is related to the ITCZ location changes?

Line 182-185, what is the time period/frequency when considering the ITCZ latitude and the latitude of Ψ_{max} ? Looks like the orange shading is shared between LIA and MCA. Are you sure the ascending branch locates at the same position between these two periods?

Reviewer #3 (Remarks on code availability):

I am traveling in a place with restricted access to certain websites currently. I don't know if it's the firewall or the website itself is blocking me from opening it.

Reviewer 1

General comments:

The revised version of the paper is improved in comparison to the original version. An extra effort has been performed to additionally analyse three CMIP6 runs and to add realistic (RCP4.5 and SSP2-4.5) emission scenarios. While I think the results are based on an extensive analysis that in general merits a publication in Nature Communications, many claims (even in the Abstract and in the Title) are not unambiguously backed by the results. Some statistical analyses in the paper are inadequate and should be revised. The sensitivity of the trends to the applied time interval should be performed. Therefore, I would suggest a major revision of the paper.

We thank the reviewer for providing valuable feedback. In addition to answering all the reviewer's comments from the first round, we have addressed the reviewer's new comments below.

Major comments:

1. Abstract, lines 12-13: The authors state "natural forcings acted to intensify the circulation". I fail to fully see how this claim is backed by the results looking at Figures 1a and Extended Data Figure 11a. For example, looking at Figure 1a, the trendline difference between 850 and 1849 seems:

- a. Smaller than the standard-deviation of the ensemble-mean time series
- b. Much smaller than the ensemble spread.

Saying this, I am yet not fully convinced by the statistical approach applied in this study to test the significance of the trend in 850-1849. Instead of evaluating the trend for each model run and then performing Student-t test for the mean of the trends (as the Authors did in Figure 1b and Extended Data Figure 11b), one would have to test significance of the trend using Mann Kendall test. The Mann-Kendall statistical test for trend is used to assess whether a set of data values is increasing over time or decreasing over time, and whether the trend in either direction is statistically significant. One should therefore:

- a. Apply Mann-Kendall test to every model run (is each of the trends significant between 850 and 1849? I suspect they are not because the internal variability signal has larger magnitude)
- b. Apply Mann-Kendall test to estimate the significance of the trend of the ensemble mean of the runs (in lines 72-73, the Authors even say, that they focus on the ensemble mean)

Note that the significance of the trend in Mann-Kendall test depends mostly on the magnitude of the trend in comparison to the standard deviation of the time series, which implicitly evaluates whether and to what extent the evaluated trend is dependent on the applied time interval. For example, would the trends be the same if you apply the starting interval at year 950 instead of 850? What about changing the end interval to 1750 instead of 1850. Would the conclusions stay the same? Are the results robust to the alteration of the start/end year or both? This would also have to be tested with respect to the analysis described in lines 111-117, and the analysis should be performed for Ψ_{avg} too. See for example Chung et al., 2019, their Fig. 2.

Considering that, it remains questionable whether the title is correct (“reverse a multi-century naturally forced intensification”). It might well be a stagnation. It also remains questionable whether the language in the remaining of the Abstract is correct. The text should therefore be modified according to the outcome of the Mann-Kendall statistical test and sensitivity tests of the applied interval for trend evaluation. For an independent review, the time series used to generate Figures 1a and Extended Data Figure 11a should also be appended in the revised version. To me, besides Ψ_{avg} for CESM, the trends do not seem significant. They seem susceptible to the applied interval.

Following the reviewer’s comment, we now show that our results are robust even under-implemented the Mann-Kendall test (lines 95-98, 326-330 and Extended Data Fig. 16) and under different trend intervals (lines 98-99, 330-333, and Extended Data Fig. 17).

First, in response to the reviewer’s request, Fig. 1 below illustrates the temporal evolution of individual members/models (CESM in orange and CMIP5 in grey) used to generate CESM and CMIP5 means (red and black lines, respectively). Note that Fig. 1a is for the Ψ_{max} metric and Fig. 1b is for Ψ_{avg} . As can be seen, distinguishing individual time series becomes challenging because of the time series length. Moreover, comparisons between Fig. 1a below and Fig. 1a in the manuscript and Fig. 1b below and Extended Data Fig. 13a indicate striking similarities. Hence, the shading representing standard deviation across members/models effectively captures the models’ spread, rendering the presentation of individual members’/models’ evolution redundant for our conclusions.

Second, the CESM and CMIP5 mean have additional and great meanings besides the statistical definition. In the climate modeling scientific field, averaging a sufficiently large ensemble of climate simulations allows referring to the forced response of the system by filtering out the internal variability processes. Note that for a multi-model ensemble, the ensemble mean still includes the variability that stems from the inherent difference between the models, in contrast to the mean of a single-model ensemble. Therefore, our study, which focused on understanding the system’s forced response through climate model simulations, primarily relies on ensemble means. Individual model analyses become less pertinent as they retain internal variability, potentially obscuring forced response variations. Our conclusion regarding the unprecedented nature of the NH HC’s present and future human-induced weakening, contrasting with naturally

forced strengthening over the last millennium, predominantly stems from analyses of CESM and CMIP5 ensemble means. Specifically, Fig. 2 below shows that the forced response strengthening trends remain robust across different time intervals (ranging from 600 to 1000 years), with a consensus among individual members/models on the sign of the trend. Furthermore, the year of emergence from the internal variability (as shown in the Manuscript's Fig. 2b) remains consistent across various time intervals, as one can see from Fig. 3 below; all different time interval linear trends emerge between years 1600-1650. Thus, the above sensitivity tests demonstrate the robustness of our results.

Finally, we adopted a more comprehensive approach by incorporating the Mann-Kendall test (Hussain and Ishtiaq, 2019) to assess the significance of NH HC last millennium strengthening trends across individual members/models and ensembles' means (Fig. 4 below, lines 323-328 and Extended Data Fig. 16). Originally, we employed linear regression to calculate trend values, followed by significance testing using confidence intervals, focusing primarily on ensemble means to discern forced response NH HC strengthening (lines 95-98). Fig. 1b in the manuscript highlights that despite considerable variability between individual runs, the ensemble mean with the confidence intervals consistently depicts a strengthening trend. The results of the Mann-Kendall test are shown in Fig. 4 below, in panels a for Ψ_{\max} and b for Ψ_{avg} . It can be seen that, in general, the resulting Mann-Kendall values of the slopes are similar to the original, both in the individual runs and in the ensemble's mean, and have positive values (except for one) (compare panel a to Fig. 1b in the manuscript and panel b to Extended Data Fig. 13b). Moreover, the ensemble mean (i.e., the forced response of the HC) also shows a significant intensification under the Mann-Kendall test, further corroborating our results. We note that although some models have insignificant trends, the forced response (which we focus on in our research) is significant. In other words, while internal variability could result in insignificant trends in some of the models (as also occurred under human-induced changes in recent decades (Deser et al., 2012, 2016)), the mean is significant; the effect of natural forcings on the flow is robust. As discussed previously, our focus on ensemble means aligns with our objective of understanding the NH HC's response to natural forcings. Additionally, our physical explanations, supported by multiple analyses, further reinforce the conclusion of NH HC strengthening over the last millennium, linking it to the cooling between the MCA and LIA, resulting in decreased static stability and increased meridional gradient of latent heating in low latitudes.

To summarize, it is crucial to emphasize the importance of examining the forced response of the system through the ensemble mean when studying the response of the NH HC to external forcings across historical and future periods. While the reviewer suggested additional analyses, it is important to note that our focus on the ensemble mean remains essential for capturing the overarching trends and filtering out internal variability.

Our results underscore the robustness of the forced response, particularly evident in the significant strengthening of the NH HC in the ensemble mean despite potential variability in individual model runs. Thus, our conclusions in the abstract and title hold.

Figure 1: Evolution of **a**, Ψ_{\max} and **b**, Ψ_{avg} , relative to the 1810-1850 period, in CESM mean (red line) and in CMIP5 mean (black line). Thin lines show the evolution of individual members/models. The evolution has been smoothed with a 3-year running mean for plotting purposes.

Figure 2: Trends ($\text{kgs}^{-1}\text{yr}^{-1}$) in the Ψ_{\max} (**a-b**) and the Ψ_{avg} (**c-d**) for the ensemble mean of CESM (**a,c**) and CMIP5 (**b,d**). Black dots show where less than two-thirds of the members/models agree on the sign of the trend.

Figure 3: Years of emergence of the Ψ_{\max} out of the internal variability (Methods).

Figure 4: The 850-1849 **a**, Ψ_{\max} and **b**, Ψ_{avg} trends in CESM (red) and CMIP5 (black) based on the Mann-Kendall test (Methods). The red and black dots show the CESM and CMIP5 mean trends, respectively, and the crosses (pluses) show the individual models' trends with p-values less (more) than 0.05. Note that the mean trend for both ensembles is significant.

2. Abstract lines 15-16. In any case, the effect of natural forcings during the 850-1850 does not seem to be large enough to vastly alter the Hadley cell strength. The language in the Abstract (“the large effect of natural forcings on the Hadley circulaion”) should thus be more modest, e.g. “the amplifying effect of natural forcings on the Hadley circulation”. The same applies to Discussion section in lines 247-248.

Following the reviewer’s comment, we have modified the above lines (lines 15-16 and 256-258). We would like to stress, however, that, first, the multi-centennial HC intensification over the last millennium is evident in both ensemble’s means (i.e., the forced response of CESM members and CMIP5 models) and found to be significant under multiple tests, e.g., Student-t (Fig. 1b in the manuscript) and Mann-Kendall (Extended Data Fig. 16) tests. We want to note again that we are interested only in the forced response to screen out the impact of internal variability on the HC strength, as was done in numerous previous studies (Deser et al., 2012; Santer et al., 2013a,b; Deser et al., 2016). Second, using single-forcing ensembles, we conducted attribution analysis and revealed that the intensification of the circulation in past centuries is mostly due to natural forcings (Extended Data Fig. 4). Third, the signal-to-noise analysis result

suggests that the emergence of the intensification, in both CESM and CMIP5 means, out of the internal climate variability can be mainly attributed to natural forcings due to the evidence that internal variability alone could not adequately explain the NH HC intensification over the last millennium. Moreover, the emergence of all CESM members and almost all CMIP5 models stresses the significance of naturally forced strengthening.

We agree that the rate of the multi-centennial HC intensification over the last millennium is remarkably lower relative to the present/future weakening of the NH HC, which strengthens our statement regarding the unprecedented nature of the anthropogenically forced weakening of the circulation. However, using multiple analyses, we revealed that the ability of natural forces to impact the circulation is not negligible; natural forcings are the dominant factor for the NH HC intensification over the last millennium.

3. The title. The results on NH Hadley cell intensification are not backed by observed data, e.g. paleoclimate data revealing the surface ocean currents in the tropical/subtropical oceans. Therefore, the title should unequivocally suggest that the outcome of the study is based on climate model simulations and is therefore not an undisputable fact, despite using reconstructions of external forcings. This has already been suggested in the previous review. I would suggest sth. like: “Anthropogenic forcings reverse a [simulated] multi-century naturally-forced Northern Hemisphere Hadley cell intensification.” If the authors want to state it as a fact, they should perform a more extensive study using paleoclimate data.

We thank the reviewer’s comment regarding the manuscript’s title. Accordingly, as suggested by the reviewer, we revised the title to include explicitly the fact that we use climate model simulation to investigate the NH HC strength over the last millennium. As mentioned in our previous review’s answer, the use of the term ‘naturally-forced’ in the title should imply the forced response of the HC to external forcings, i.e., model-based results. However, we recognize that this terminology may confuse readers outside the scientific field. Therefore, adding the word ‘simulated’ to the manuscript’s title aims to enhance the accuracy and understandability of our main result for a broader audience.

Note that while we acknowledge the importance of paleoclimate data in complementing model-based studies of the last millennium, it’s crucial to recognize the limitations inherent in such data sources. Proxy data, while valuable, often have spatial and temporal limitations that make it challenging to infer global-scale phenomena, especially wind patterns, which are critical for studying HC dynamics. On the other hand, state-of-the-art climate models offer comprehensive and globally consistent simulations of atmospheric circulation patterns. By integrating these model simulations alongside reconstructions of external forcings, we can better understand the mechanisms driving changes in the NH HC intensity over the last millennium in response to natural forcings.

Lines 201-206: The arguments presented in this paper are not sufficient to call it a positive feedback loop. First, the KE equation is diagnostic, and it only describes that the diabatic heating meridional gradient mostly changes proportionally with the Laplacian of the stream-function. KE equation does not describe any lead-lag relationship between the two, but obviously the two change in accordance, and this is captured by the KE equation. Second, the change in the mean-circulation ($\bar{q}\Delta\bar{v}$) is also the change of the circulation itself. If you integrate $\Delta\bar{v}$ vertically and \bar{q} was constant vertically, you would get $\Delta\Psi$. However, \bar{q} decreases approximately exponentially with height, so what you get is the contribution to $\Delta\Psi$ from the lower HC branch, which brings the fuel for condensation, which in turn helps maintaining the meridional gradient in adiabatic heating. However, again, the moisture budget does not include any lead-lag mechanism. All this does not mean that the positive feedback loop is not actually present, it only means that the evidence presented in the manuscript is not sufficient to call it a feedback loop. Adjustment experiments using a fully coupled climate model would be needed to elucidate the mechanism.

We thank the reviewer for this important point. We acknowledge that the KE and moisture budget equations are diagnostic, limiting our ability to assert a direct causal relationship between dynamics and the diabatic heating meridional gradient. Consequently, we have moderated our claim regarding the hypothesis of a positive feedback loop between the HC and latent heating over the last millennium (lines 205-215).

Yet, please note that our analyses indicate a strong interaction between the response to natural forcings of the circulation and latent heating in past centuries, supported by two distinct lines of analysis. First, our investigation using the KE equation underscores the predominant contribution of the meridional gradient of latent heating to changes in the NH HC between the MCA and the LIA relative to other physical process components comprising the circulation response to natural forcings (Fig. 3b in the manuscript); without the contribution from latent heating, the KE equation would not capture the HC intensification. Second, employing the moisture budget equation, we demonstrate that alterations in circulation alone can account for the increase in the meridional gradient of precipitation at the NH HC ascending branch area over the last millennium. It is noteworthy that, in contrast, by the end of this century, precipitation changes in low latitudes are primarily explained by the thermodynamic component ('wet get wetter' argument by Held and Soden (2006)).

Both analyses converge on a strong relationship between changes in the NH HC and the latent heating process in low latitudes over the last millennium. This underscores the complex interplay between the mean circulation and latent heating meridional gradient.

Other comments:

1. Line 35: I would not say “as opposed to wind observations”, try to reformulate it. The trend in the poleward winds in the upper branch of the Hadley circulation is too small to be unequivocally distinguished from the analysis-minus-background standard deviation. On the other hand, the trends in the zonal easterlies are significant and larger than the observation-sampling noise.

Following the reviewer’s comment, we revised the above line to clarify that the surface zonal wind component is the pertinent wind product for observing and verifying the low latitude circulation trend (lines 34-39). This clarification highlights that the zonal wind component stands out from the background noise, unlike the meridional wind component, which does not offer the same distinction.

2. It would be good to see CMIP6 with Ψ_{avg} as it is done with Ψ_{max} in Extended Data Figure 3.

Done (line 273 and Extended Data Fig. 3). Overall, changing the HC strength metric does not change the results utilized by CMIP6 models (Fig. 5 below); Ψ_{avg} also shows that the human-induced HC weakening in recent and coming decades is unprecedented compared to the HC forced changes over the last millennium (panel a). It also yields a HC strengthening over the last millennium at a rate of $1 \times 10^6 \text{ kgs}^{-1}\text{yr}^{-1}$ in CMIP6 mean (panel b).

Figure 5: **a**, Evolution of the Ψ_{avg} (the averaged streamfunction between 10° and 25° and between 1,000 mb and 100 mb), relative to the 1810-1850 period, in CMIP6 mean. Shading shows s.d. across models. **b**, The 850-1849 Ψ_{avg} trends in CMIP6. The black dot shows the mean trend, and the crosses show the individual models’ trends. The error bar shows the 95% confidence interval of the mean trend based on a Student’s t-distribution.

3. Lines 58-61: The authors first state, that the changes in volcanic forcings are not considered in climate model projections. What do the Authors then mean by “would allow one to better constrain the future climate changes”. First, do you mean projections of future climate change? Second, how could the projections be constrained? For example we do not know, when some volcano is going to erupt, how much material will it deposit into the atmosphere, what will be the emission altitude, what would be the ratio between warming constituents (e.g. stratospheric water vapour) and cooling constituents (e.g. sulfate aerosols). Perhaps, the Authors had some other natural forcing in mind, which could actually better constrain the projections. I suggest some reformulation of the text.

We thank the reviewer for this useful comment, and we reformulated the above lines (lines 57-61) accordingly. First, since volcanic eruptions are not well incorporated in model projections (they hold constant values in both CMIP5 and CMIP6), better understanding their impacts on the flow would allow one to constrain model projections. Specifically, such eruptions’ stochastic and unpredictable nature challenges their seamless incorporation into prescribed forcings for future model scenarios. Chim et al. (2023) conducted climate simulations spanning from 2015 to the end of the century, incorporating stochastic future eruption scenarios to assess the impact of volcanic forcing uncertainties on climate projections. Their findings revealed substantial underestimation in current model scenarios compared to median future stochastic scenarios, particularly concerning effects on large-scale climate indicators such as surface temperature. They advocate for using stochastic eruption scenarios generated with state-of-the-art volcanic emission inventories or time-averaged constant forcing derived from such scenarios to better capture long-term volcanic activity.

The flawed representation of volcanic eruptions in current climate projections doubts the accuracy of predictions of future climate. To enhance the precision of these projections and effectively constrain model projections, it is imperative to deepen our understanding of the climatic ramifications of volcanic forcings and incorporate them more accurately into model projections.

4. 148-149 and Figure 3: the methodology how the authors obtained the contribution of Q_{lat} and Q_{rad} from Q , obtained by KE decomposition, is not clear. Were Q_{rad} and Q_{lat} available as model outputs? Was column-integrated Q_{lat} estimated from precipitation? Describe the methodology.

According to the reviewer’s comment, we now explicitly mention that the diabatic heating (and also the latent and radiative heating that comprise it) are available as model outputs in the CESM model (lines 152-153 and 383-384). Thus, it is important to note that the net latent heating in the atmospheric column is also directly available as a model output. Therefore, we did not need to calculate one from the other; both were directly obtained from the model outputs.

5. “gravity” instead of “gravity acceleration” would suffice. I used “gravity acceleration = gravitational acceleration + centrifugal force” only to demonstrate the difference between the terms gravitational acceleration and gravity.

Done (line 267).

6. Line 192: I would recommend using the same form of equation as in Eq.6. While the Authors righouly say the equation is the vertically integrated moisture budget equation, the ∇_y went missing here neither is mentioned.

Done (lines 192-196).

7. 223-225: NH and SH exhibit similar changes in temperature only in the tropics and subtropics (which mader mostly for the HC). Please reformulate accordingly.

Done (lines 233-236).

8. Lines 261-262: I am not sure “similar” is the right expression. The emergence of the trend out of the noise is not similar using the average cell strength Ψ_{avg} as it appears later than as in the case of Ψ_{max} (2010-2020), right?

Analyzing Ψ_{avg} yields similar results in the sense that the human-induced HC weakening in recent and coming decades is unprecedented compared to the HC’s forced changes over the last millennium. This contrasts with the HC strengthening induced by natural forcings during the same period. Re-calculating the emergence of the HC weakening using the Ψ_{avg} metric yields similar results of emergence, around 2020, out of the last millennium forced NH HC changes (Fig. 6 below, lines 273, and Extended Data Fig. 14).

Figure 6: Signal-to-noise ratio analysis to the Ψ_{avg} trend from 1970 and to each year plotted against the last year of trend in CESM mean (red line) and CMIP5 mean (black line). Shading shows the s.d. of signal-to-noise ratio values (Methods). The horizontal black line represents a signal-to-noise ratio value of -2.

9. Lines 261-262: In response to Reviewers' comments, the Authors state: "Specifically, we follow Pikovnik et al. (2022) and average the meridional mass streamfunction between 10° and 25° and between 1,000 mb and 100 mb (Ψ_{avg} , Fig.2 below)." This is not the definition of the average Hadley cell strength defined in Pikovnik et al. (2022), see their Eq. 3. Please, revise accordingly.

Thank you for bringing this to our attention. While we acknowledge the reference to Pikovnik et al. (2022), we opted to define our study's average HC strength differently, specifically focusing on the annual mean NH HC over the last millennium. Our choice to average the meridional mass streamfunction between 10° and 25° and between 1,000 mb and 100 mb (Ψ_{avg}) was deliberate, aiming to exclude any potential influence from the SH HC or flow beyond the HC boundaries. Although the definition slightly differs from Pikovnik et al. (2022), we find it suitable for our study's objectives and maintain its relevance to our annual mean NH HC analysis.

10. Lines 288-301: which variables did the Authors obtain? As far as I understand from Fig 3b, Q_{lat} and Q_{rad} were available?

This is correct; the heating components (Q , Q_{lat} and Q_{rad}), as well as the eddy fluxes, precipitation, meridional velocity, and temperature, are available CESM model outputs.

11. 362: define how Q was computed – as a thermodynamic residual?

As mentioned above, Q is an available CESM model output.

12. 364: Running or static monthly-mean?

The static monthly mean data is an available output of the model simulations.

13. 385-386: Same question: are the eddies computed as deviation from a static or running annual-mean zonal-mean

The eddies are computed as deviation from static annual-mean zonal-mean.

Reviewer 3

This revised paper focused on the impact of natural forcings on Northern Hemisphere (NH) Hadley Cell (HC) over the last millennium. This time, CMIP6 simulations were also analyzed. This version firstly stated that, over the last millennium, it is the natural forcings that strengthen the NH HC instead of internal variability and human-induced forcings. To find the source of the natural variability, this study again used KE equation to decompose the HC strength to different natural forcings and found that static stability and latent heat are the two major factors. My suggestion is to accept the paper with major revisions. The reasons are as follows:

We thank the reviewer for the careful reading and very useful comments. In addition to answering all reviewer comments from the first round, we address the reviewer's new comments below.

This is a good idea to examine the ITCZ responses per other reviewers' request. However, I feel the section "Linking the Hadley cell strengthening to the cooling over the last millennium" needs a more comprehensive explanation. Currently, the static stability's role is well explained but not the latent heat part.

Firstly, for the location changes of the ITCZ, can you explain clearer on how the HC intensity would affect the location/shift of the ITCZ? I am also wondering if it is the location of the ITCZ or the width of the ITCZ is more related to the intensity of the HC.

Following the reviewer's comment, we have expanded our analysis to encompass not only the shifts in the location but also the variations in the width of the ITCZ over the last millennium (lines 230-236 and Extended Data Fig. 9). As suggested by another reviewer, a possible explanation for the NH HC strengthening over the last millennium, beyond what we are arguing, is uneven heating between the hemispheres; changes in the HC strength might be expected due to more energy transport from one hemisphere to another. As the HC transports moist static energy in the direction of the flow in its upper branch, the energy is transported toward the cooled hemisphere, accompanying an ITCZ shift towards the warmed hemisphere (Kang, 2020). In addition, it is conceivable that a shifted ITCZ might alter the meridional gradient of precipitation (or latent heating), thus affecting the HC strength. However, further investigation of the evolution over 850-2100 of the ITCZ location (ϕ_{ITCZ} , defined as the latitude where the 500 mb meridional mass streamfunction, Ψ , is zero) and width (W_{ITCZ} , defined as the distance in degrees latitude between the ITCZ edges, which are the latitudes closest to the equator at which the meridional derivative of the 500 mb meridional mass streamfunction, $\frac{\partial \Psi}{\partial \phi}$, is zero (Byrne et al., 2018)) reveal that there was no clear ITCZ shifting trend over the last millennium (shown in Extended Data Fig. 8 and Fig. 7 below, respectively). This finding, together with the hemispheric symmetric response of the low latitudes troposphere's temperature between the MCA and LIA (Fig. 4a in the manuscript and Extended Data Fig. 6a), cast doubt on the above mechanism as being responsible for the strengthening of the NH HC over the last millennium.

Instead, based on the results from the KE and moisture budget equations analyses, we suggest that over the last millennium, the intensification of the NH HC might have acted as positive feedback; the Ψ_{\max} intensification (which may be triggered by a decrease in static stability) increased latent heat release over the ascending branch of the NH HC, resulting in a larger meridional gradient of latent heating, which further strengthened Ψ_{\max} .

Figure 7: **a**, Evolution of the W_{ITCZ} (defined as the distance in degrees latitude between the ITCZ edges, which are the latitudes closest to the equator at which the meridional derivative of the 500 mb meridional mass streamfunction, $\frac{\partial \Psi}{\partial \phi}$, is zero), relative to the 1810-1850 period, in CEM5 mean (red line) and CMIP5 mean (black line). Shading shows s.d. across members/models. **b**, The 850-1849 W_{ITCZ} trends in CEM5 (red) and CMIP5 (black). The red and black dots show the CEM5 and CMIP5 mean trends, respectively, and the crosses show the individual members/models. Error bars show the 95% confidence interval of the mean trend based on a Student's t-distribution.

I can see the latent heat intensifies the NH HC but how it is related to the ITCZ location changes?

In response to the reviewer's comment, we have further elaborated on how changes in the ITCZ location can modulate the meridional gradient of latent heating and subsequently influence the intensity of the HC (lines 230-236). Specifically, as mentioned above, a shift of the ITCZ would change the gradient of net latent heating, which affects the HC strength.

We added this possible mechanism to our physical explanation due to another reviewer's suggestion. However, despite the potential significance of this mechanism, our analysis does not yield sufficient evidence supporting a substantial ITCZ shift over the last millennium period. Additionally, our investigation did not uncover significant disparities in heating between the hemispheres. While we acknowledge the importance of considering this mechanism as a potential explanation for HC strengthening, the lack of compelling evidence suggests it may not be a primary driver in this context.

Line 182-185, what is the time period/frequency when considering the ITCZ latitude and the latitude of Ψ_{\max} ? Looks like the orange shading is shared between LIA and MCA. Are you sure the ascending branch locates at the same position between these two periods?

Following the reviewer's comment, we have explicitly clarified in the manuscript text that the location of the NH HC ascending branch is determined based on the MCA climatology (lines 184-185) and show that this location doesn't change between the MCA and LIA (Fig. 8 below). This choice is grounded in the understanding that precipitation and latent heating changes are highly dependent on the climatological state, as was shown by Held and Soden (2006). Therefore, focusing on the deep tropics, where the climatological state corresponds to a moisture flux convergence equivalent to the ascending branch of the HC, allows for a more accurate assessment of changes in meridional gradients of latent heating and precipitation. Note that we do not define the ascending branch over the other side of the ITCZ to avoid including processes irrelevant to the changes in the annual NH HC. Furthermore, the use of MCA values allows us to explore the changes in precipitation in the context of the wet-get-wetter mechanism (Held and Soden, 2006). In climatological regions (i.e., during the MCA) of moisture flux convergence, one would expect a decrease in precipitation under surface cooling. Here, however, the precipitation increases due to dynamical processes.

Figure 8: The NH HC ascending branch's boundaries; the black (blue dashed) lines represent the MCA (LIA) period boundaries.

Remarks on code availability

I am traveling in a place with restricted access to certain websites currently. I don't know if it's the firewall or the website itself is blocking me from opening it.

We've tested the DOI link provided, and it appears to work on our end.

References

- Md. Hussain and M. Ishtiak. pyMannKendall: a Python Package for non Parametric Mann Kendall Family of Trend Tests. *J. Open Source Softw.*, 4(39):1556, 2019.
- C. Deser, A. Phillips, V. Bourdette, and H. Teng. Uncertainty in Climate Change Projections: the Role of Internal Variability. *Clim. Dyn.*, 38:527–546, 2012.
- C. Deser, L. Terray, and A. S. Phillips. Forced and Internal Components of Winter Air Temperature Trends over North America during the past 50 Years: Mechanisms and Implications. *J. Clim.*, 29:2237–2258, 2016.
- B. D. Santer et al. Identifying Human Influences on Atmospheric Temperature. *Proc. Natl. Acad. Sci. U.S.A.*, 110(1):26–33, 2013a.
- B. D. Santer et al. Human and Natural Influences on the Changing Thermal Structure of the Atmosphere. *Proc. Natl. Acad. Sci. U.S.A.*, 110(43):17235–17240, 2013b.
- I. M. Held and B. J. Soden. Robust Responses of the Hydrological Cycle to Global Warming. *J. Clim.*, 19:5686–5699, 2006.
- M. M. Chim et al. Climate Projections Very Likely Underestimate Future Volcanic Forcing and Its Climatic Effects. *Geophys. Res. Lett.*, 50(12):e2023GL103743, 2023.
- M. Pikovnik, Ž. Zaplotnik, L. Boljka, and N. Žagar. Metrics of the Hadley Circulation Strength and Associated Circulation Trends. *Weather Clim. Dynam*, 3(2):625–644, 2022.
- S. M. Kang. Extratropical Influence on the Tropical Rainfall Distribution. *Curr. Clim. Change Rep.*, 6:24–36, 2020.
- M. P. Byrne, A. G. Pendergrass, and A. D. Rapp et al. Response of the Intertropical Convergence Zone to Climate Change: Location, Width, and Strength. *Curr. Clim. Change Rep.*, 4: 355–370, 2018.

REVIEWERS' COMMENTS

Reviewer #1 (Remarks to the Author):

The authors have adequately addressed all my concerns and provided excellent responses. I would suggest the acceptance of the paper for Nature Communications.

Reviewer #3 (Remarks to the Author):

Compared to the original version, this paper concentrated on the model simulated Northern Hemisphere Hadley cell variations over the last millennium which helps the paper's arguments more focused. A common problem that almost all model simulation studies facing is that how much should we trust the model simulations. As you may find, many of the energetic theory studies are based on more idealized models. I would not be too surprised to see the coupled model simulations and idealized model simulations are different. It's still worthy to report what the coupled models simulated and not simulated. Therefore, this paper is suitable for publication. However, I do have a minor question about the title, currently my first reaction to the title is that this paper is going to talk about how and how much anthropogenic forcings reverse the simulated NH HC intensification which is not the main argument of this paper.

Other thoughts (no worries, not comments):

A trend analysis can be hard/tricky because it depends on your time periods. I do agree with other reviewers that a student t test is not a very good choice in general for most significant tests since it assumes the data follow a Gaussian distribution.

Reviewer #3 (Remarks on code availability):

The code is for KE equation calculation in python script that can be used for other similar format data.

Reply to reviewers comments:

Anthropogenic forcings reverse a simulated multi-century naturally-forced Northern Hemisphere Hadley cell intensification

Reviewer 1

The authors have adequately addressed all my concerns and provided excellent responses. I would suggest the acceptance of the paper for Nature Communications.

We appreciate the reviewer's thorough examination and insightful feedback.

Reviewer 3

Compared to the original version, this paper concentrated on the model simulated Northern Hemisphere Hadley cell variations over the last millennium which helps the paper's arguments more focused. A common problem that almost all model simulation studies facing is that how much should we trust the model simulations. As you may find, many of the energetic theory studies are based on more idealized models. I would not be too surprised to see the coupled model simulations and idealized model simulations are different. It's still worthy to report what the coupled models simulated and not simulated. Therefore, this paper is suitable for publication. However, I do have a minor question about the title, currently my first reaction to the title is that this paper is going to talk about how and how much anthropogenic forcings reverse the simulated NH HC intensification which is not the main argument of this paper.

We appreciate the reviewer's thorough examination and insightful feedback. Firstly, we wish to emphasize that throughout all the paper's versions, our focus remains on the forced response of the NH HC intensity to external forcings over the last millennium, simulated by fully coupled models. Second, we concur with the reviewer on the importance of leveraging the knowledge we can obtain from today's state-of-the-art climate models, especially if they divert from idealized simulations. Overall, the utilization of physical simulations, particularly those derived from state-of-the-art fully coupled models, serves as a cornerstone for advancing our understanding of Earth's climate system and atmospheric dynamics. These models effectively capture the system's complexity, enabling us to comprehensively explore its intricacies and dynamics. Our analysis actually shows that using basic arguments, as revealed by the KE equation, one could explain how natural forcings intensified the flow over the last millennium.

Finally, regarding the reviewer's concerns about the paper's title, we would like to emphasize that most of the paper's first section is dedicated to quantifying how unprecedented the NH HC weakening in response to anthropogenic forcing relative to the naturally forced NH HC changes over the last millennium. Specifically, we used a signal-to-noise ratio approach to test if and when the human-induced NH HC weakening emerges from the intensity changes induced by

natural forcings in past centuries. Subsequent sections then elucidate the underlying physical mechanisms behind the naturally forced strengthening of the NH HC over the last millennium (which was also found to be significant using a similar approach of signal-to-noise analysis). It is essential to note that while our paper does not delve into the specific mechanisms driving the human-induced weakening in present and future decades, ample research has explored this aspect, as acknowledged in our introduction.

Together, the paper's sections provide the reader with an insightful comprehension that relative to the last millennium, the human-induced NH HC weakening in present and future decades is not only unprecedented but also in contrast to the forced response strengthening by natural forcings during past centuries. Thus, the title accurately encapsulates the primary conclusion of our study.

Other thoughts (no worries, not comments):

A trend analysis can be hard/tricky because it depends on your time periods. I do agree with other reviewers that a student t test is not a very good choice in general for most significant tests since it assumes the data follow a Gaussian distribution.

We appreciate the reviewer's comment. While we understand concerns regarding using the Student-T test and its underlying assumptions, it is worth noting that this test is widely accepted in the field for analyzing the significance of a signal, such as a general trend in the time series. Additionally, we conducted sensitivity analyses using alternative methods, such as the Mann-Kendall test, confirming the intensification trend's significance (Supplementary Fig. 16). Importantly, our findings remained consistent across different intervals of trends chosen for analysis, indicating the robustness of our results (Supplementary Fig. 17).

Remarks on code availability:

The code is for KE equation calculation in python script that can be used for other similar format data.

Indeed.